# An insulin receptor activity surge in follicle cells drives vitellogenesis by upregulating CrebA

Xiaoya Wang [1,2,3,6], Huanju Liu [2,4,6], Zhiyong Yin[2,5], Tianning Shao[1,2,3], Lin Li[1,2,3], Jun Ma [1,2,3 ✉] & Feng He [1,2,3 ✉]

## Abstract

**Folliculogenesis is a process that requires accurate interpretation of female physiological cues and elaborate coordination between the growing oocyte and its surrounding follicle cells, each being capable of responding to external signals. Here, we investigate the role of insulin signaling in *Drosophila* follicle cells. Using a phase separation-based reporter system, we observe a surge of insulin receptor activity in follicle cells during vitellogenic stages, a surge that is disrupted by a maternal high-sucrose diet. Single-cell RNA-seq reveals a diet-sensitive subpopulation of stage-8 follicle cells, which exhibits a reduction in CrebA-mediated transcription of genes for yolk and vitelline membrane proteins. Our results suggest a critical role of CrebA in implementing the stage-specific effect of insulin signaling to boost the secretory capacity of follicle cells. Mechanistically, *CrebA* is directly repressed by nuclear FoxO that is subject to insulin control, a regulatory axis that we show is conserved in human granulosa cells. This study delineates a mechanism through which insulin and nutrient cues act on a developmental transition via modulating the biosynthetic and secretory functions of the ovary.**

**Keywords** Folliculogenesis; Insulin; High-Sucrose Diet; Cyclic-AMP Response Element Binding Transcription Factor
**Subject Categories** Chromatin, Transcription & Genomics; Development; Signal Transduction

## Introduction

During ovarian development, follicle cells undergo proliferation and differentiation to execute their essential roles in folliculogenesis, including epithelial formation, secretion of signaling molecules, oocyte support, and ovulation facilitation (Ma, He et al, 2016; Von Stetina and Orr-Weaver, 2011). The core of these processes, which include a multitude of signaling and gene regulation mechanisms, is likely evolutionarily conserved as exemplified by the insulin/IGF-1 signaling (IIS) pathway (Das and Dobens, 2015; Ebberink, Smit et al, 1989; Owusu-Ansah and Perrimon, 2014; Wang, Tulina et al, 2007). The *Drosophila* ovary provides a powerful model for delineating the critical mechanisms regulating oogenesis. Here, individual egg chambers are composed of a single layer of follicle cells enveloping germline cells (one oocyte and 15 nurse cells), and they progress through defined developmental stages (germarium to stage 14) inside the ovarioles, offering a tractable system to explore endocrine and dietary impacts on reproduction (Liu et al, 2022).

The activity of IIS is tightly regulated to maintain metabolic homeostasis across tissues, including individual ovarian follicles. Dysregulation of this pathway has been implicated in reproductive disorders such as polycystic ovary syndrome (PCOS), where insulin resistance (IR) and compensatory hyperinsulinemia are common hallmarks (Liu et al, 2022). More generally, excess energy intake from high-sucrose diets (HSD) and high-fat diets (HFD) is strongly associated with these conditions in humans and *Drosophila* (Morris, Coogan et al, 2012; Schwartz, Figlewicz et al, 1992). However, IIS exhibits context-dependent roles due to diverse ligand-receptor interactions and different cellular milieus (Brogiolo, Stocker et al, 2001; Hwa, Oh et al, 1999; Mohan and Baylink, 2002; Wu and Brown, 2006). In *Drosophila*, the binding between neural-derived insulin-like peptides (ILPs) and insulin receptor (InR) has been shown to act directly on germline stem cells to promote their proliferation, whereas follicle cells appear to lack early IIS dependence until stage 8 (Drummond-Barbosa and Spradling, 2001; LaFever and Drummond-Barbosa, 2005). Paradoxically, at later stages when vitellogenesis takes place, mutations in the IIS pathway can affect follicle cells, resulting in reduced proliferation, increased apoptosis, and paused mitosis-to-endocycle switch (Jouandin, Ghiglione et al, 2014; LaFever and Drummond-Barbosa, 2005; LaFever, Feoktistov et al, 2010), suggesting a role of IIS in follicle cells. However, it remains to be resolved precisely how follicle cells, through their stage-specific intracellular events and dynamics, implement the effects of IIS toward successful ovulation.

IIS orchestrates metabolic and developmental programs via many conserved transcription factors, including Forkhead Box O (FoxO) (Haeusler, McGraw et al, 2018). However, relatively little is

[1]Department of Obstetrics and Gynecology, the Fourth Affiliated Hospital of School of Medicine, Zhejiang University, Yiwu, Zhejiang 322000, China. [2]Institute of Genetics, International School of Medicine, Zhejiang University, Yiwu, Zhejiang 322000, China. [3]Center for Genetic Medicine, International Institutes of Medicine, Zhejiang University, Yiwu, Zhejiang 322000, China. [4]Present address: Department of Medical Genetics, Naval Medical University, Shanghai 200433, China. [5]Present address: Department of Molecular and Human Genetics, Huffington Center on Aging, Baylor College of Medicine, Houston, TX 77030, USA. [6]These authors contributed equally: Xiaoya Wang, Huanju Liu. ✉E-mail: jun_ma@zju.edu.cn; feng_he@zju.edu.cn

known about insulin-responsive transcription factors in ovarian follicle cells. In this study, we identify cyclic-AMP response element binding transcription factor A (CrebA) as a key mediator of InR activity in *Drosophila* follicle cells during vitellogenesis. CrebA supports the biosynthetic function of follicle cells to secrete yolk and vitelline membrane proteins, which are critical for oocyte maturation. In female flies that develop IR under a high-sucrose diet, FoxO is translocated into the nuclei and inhibits *CrebA* transcription at vitellogenic stages. The InR-FoxO-CrebA regulatory axis that we identify in follicle cells is conserved in human granulosa cells as CREB3L2, a CrebA ortholog, responds to FOXO1 and has a role in regulating estradiol secretion. Our findings thus delineate an evolutionarily conserved mechanism whereby CrebA/CREB3L2 integrates IIS with reproductive physiology, offering insights into metabolic disruptions in human disorders such as PCOS.

# Results

## InR activity in follicle cells peaks at stages 8–10 and is diminished by HSD

To monitor InR activity in single-follicle cells across folliculogenesis, we used *tj-Gal4* to drive the expression of InR[SPARK], a phase separation-based EGFP reporter to probe the kinase activity of InR (Li, Dong et al, 2022). Prior to stage 8, the EGFP droplet was largely undetectable (Fig. 1A), in agreement with a lack of cell-autonomous role of InR in follicle cells at previtellogenic stages (LaFever and Drummond-Barbosa, 2005). During stages 8–12, EGFP droplets became abundant in main-body follicle cells (Figs. 1A,B and EV1 for analysis of stretched cells and centripetal cells). Quantification of the droplet-to-cytoplasm ratio showed that the InR[SPARK] signal in main-body follicle cells peaked at vitellogenic stages 8~10, followed by a sharp decline at later stages (Fig. 1B).

To verify our detection system, we evaluated the InR[SPARK] profiles in follicle cells under conditions with compromised IIS. First, we co-expressed InR[SPARK] in follicle cells with a dominant negative InR (InR[DN]; Fig. 1C), and observed a three-fold reduction in the InR[SPARK] peak level (Fig. 1D). Second, we took advantage of a diet-induced model we recently established, in which oogenesis is stalled by a maternal high-sucrose diet (HSD) that causes ovarian insulin insensitivity (Liu, Li et al, 2022). Similar to our previous finding in *w1118* females, the fraction of stage-8 egg chambers was increased in *tj > InR[SPARK]* females, from $9.5 ± 6.0\%$ under a normal diet (ND) to $26.8 ± 4.4\%$ under HSD (Fig. 1E). Consistently, the InR[SPARK] peak level was significantly lower in HSD females than that in ND females (Fig. 1F,G). Together, these results document a stage-specific surge of InR activity in vitellogenic follicle cells as detected by our SPARK reporter, a surge that is dampened under conditions of compromised IIS.

## An aberrant transcriptomic state of stage-8 follicle cells caused by HSD

To identify transcriptomic changes in response to altered IIS in vitellogenic follicle cells under HSD, we generated ovarian scRNA-seq libraries. After quality control, we recovered 14,474 and 20,592 single cells from the two libraries (ND and HSD),

respectively. The two transcriptomic datasets exhibited an excellent correlation with one another and, importantly, with both a published ovarian scRNA-seq dataset (Jevitt et al, 2020; Data ref: Jevitt et al, 2020) and our own accompanying bulk RNA datasets, indicating high reproducibility (Appendix Fig. S1). We performed unsupervised clustering analysis on our two libraries, and the result showed that all the 18 clusters were populated by cells from both conditions (Fig. 2A). Marker genes for different stages and types of ovarian cells in adult flies have been established (Jevitt et al, 2020; Rust, Byrnes et al, 2020; Slaidina, Gupta et al, 2021). Based on the fractions of expressing cells and the average expression levels of these marker genes, we assigned cell identities for all the clusters, including Malpighian tubule cells, oviduct cells, seminal receptacle cells, muscle sheath cells, germline cells, and the developmental continuum of somatic follicle cells (Figs. 2B and EV2 for the representative marker genes). Importantly, within the identified continuum of follicle cells, the two conditions exhibited differences in the distribution of specific cell stages. Notably, the fraction of stage-8 follicle cells was 1.8-fold higher in HSD than that in ND, in agreement with the oogenesis stalling at stage 8 in females under HSD.

To evaluate the transcriptomic characteristics of follicle cells at the HSD-sensitive stages, we re-clustered the 2133 follicle cells at stages 8–10 under the two conditions (Fig. 2C). Pseudotime analysis revealed two successive states that were common for both ND and HSD (states #1 and #2), and one bifurcate state that was exclusive to stage-8 HSD follicle cells (state #3). This result suggested that follicle cells from egg chambers arrested at stage 8 under HSD might have entered a transcriptomic state that is distinct from that of normal stage-8 follicle cells. To evaluate this possibility, we obtained the pseudotemporal profiles of genes whose expression levels were significantly lower in state #3 than those in state #2. These genes exhibited a dramatic increase at the branching point under ND, an increase that was dampened or reversed under HSD (Fig. 2D). Gene Ontology (GO) analysis revealed that these genes are enriched in functional categories of protein transport and vitelline membrane formation (Fig. 2E), functions known to be crucial for folliculogenesis progression through early vitellogenic stages.

## A stage-specific CrebA-mediated regulon in follicle cells revealed by the HSD model

To identify the key regulatory modules underlying the difference between the normal and HSD-induced states of follicle cells, we used SCENIC (Van de Sande et al, 2020) to infer the single-cell transcription regulatory network. This method first identified co-expression modules between transcription factors (TFs) and putative target genes. The modules with significant enrichment TF binding motifs were refined and referred to as regulons (Aibar, González-Blas et al, 2017). For each regulon in each cell, the activity was scored by assessing the expression ranks of the target genes in the transcriptome, and the score was binarized by fitting a Gaussian mixture model for further analysis. Figure EV3 plots the binary scores for all the 42 regulons predicted from stages-8–10 follicle cells. We found that 24 regulons were intensely activated in late follicle cells of HSD-induced state #3, including transcription factors that are involved in or interact with ecdysteroid signaling, such as Ecdysone receptor (EcR), Enhancer of bithorax (E(bx)),

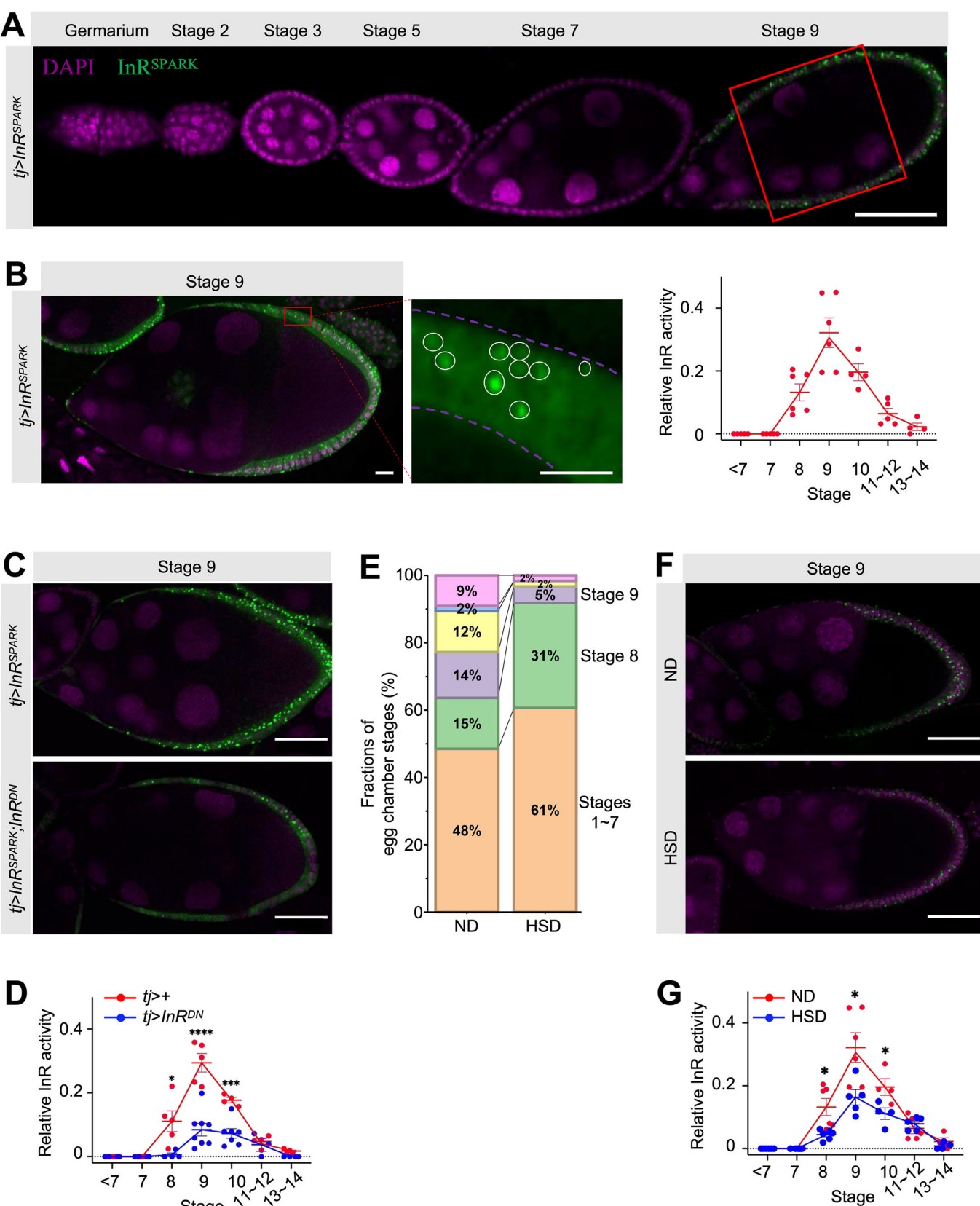

◄ **Figure 1.  InR activity in follicle cells peaks at stages 8–10.**

(A) A mid-sagittal image showing InR^SPARK signals (green) in follicle cells at different stages from *tj > InR^SPARK* females. Nuclei were counterstained with DAPI (magenta). Main-body follicle cells away from the two poles (marked by the red box) were used for quantification. Scale bar = 50 μm. (B) Left: an illustration of the method for quantifying the InR activity. Scale bar = 50 μm. From the magnified region, quantification was performed by normalizing the summed pixel intensities of all cytoplasmic droplets (within white circles) within a given region to the total intensity of the cellular region (between dashed lines). Scale bar = 10 μm. Right: quantified InR activities of main-body follicle cells at different stages. N = 6, 6, 4, and 5 egg chambers for stages 8, 9, 10, and 11–12, respectively. Error bars represent mean ± standard error of the mean (SEM). (C) Representative images of InR^SPARK signals (green) in stage-9 follicle cells from *tj > InR^SPARK* (top) and *tj > InR^SPARK;InR^DN* females (bottom). Scale bars = 50 μm. (D) Quantification of InR^SPARK signals in main-body follicle cells at different stages from *tj > +* (red; N = 5, 5, 4, and 4 egg chambers, respectively) and *tj > InR^DN* (blue; N = 4, 8, 7, and 3, respectively) females. Error bars represent mean ± SEM. Student's *t*-test *p* values = 0.03 (denoted as *), <0.0001 (****) and 0.0008 (***) at stages 8, 9, and 10, respectively. (E) Distributions of egg chambers at different stages from *tj > InR^SPARK* females under a normal diet (ND, N = 3 females) or a high-sucrose diet (HSD, N = 4 females). (F) Representative images of InR^SPARK signals (green) in stage-9 follicle cells from *tj > InR^SPARK* females under ND (top) or HSD (bottom). Scale bars = 50 μm. (G) Quantification of InR^SPARK signals in main-body follicle cells at different stages from ND (red; N = 6, 6, 4, and 5 egg chambers, respectively) and HSD (blue; N = 6, 5, 4, and 5, respectively) females. Error bars represent mean ± SEM. Student's *t*-test *p* = 0.01 (*), 0.03 (*), and 0.04 (*) at stages 8, 9, and 10, respectively. Source data are available online for this figure.

estrogen-related receptor (ERR), Jun-related antigen (Jra), and ftz transcription factor 1 (ftz-f1). These regulons are indicative of a progression towards apoptosis of HSD egg chambers arrested in stages 8–9 (Terashima and Bownes, 2006). Figure 3A shows the scores for the four HSD-deactivated regulons: slbo, Hr4, Xbp1, and CrebA. Among these four transcription factor genes, *CrebA* showed the highest temporal specificity in follicle cells of stages 9–10b, an expression that was reduced under the HSD condition (Fig. 3B). To verify this HSD-induced reduction of CrebA at the protein level, we performed immunostaining and Western blot analysis. The results show that CrebA was predominantly localized in the nuclei of vitellogenic follicle cells, and its total expression was significantly reduced in ovaries from HSD females (Fig. 3C,D). Together, these results suggested that deactivation of the CrebA-mediated regulon in follicle cells may be responsible for the follicular arrest in the ovaries of HSD females.

According to our predicted network, there were 629 target genes in the CrebA-mediated regulon. Similar to the downregulated genes in follicle cells at the HSD-induced state, these CrebA-regulon genes were also enriched in GO terms of protein transport and vitelline membrane formation (Fig. EV4A), supportive of a role of CrebA in regulating HSD-sensitive vitellogenic gene expression. These genes included those encoding yolk proteins (Yp1 and Yp2) and major early eggshell proteins (Vm26Aa and Vm26Ab), all of which had a peak expression during stages 9 ~ 10b and were sensitive to HSD (Fig. 3E). Ovarian bulk mRNA-seq comparing ND and HSD females revealed that 80 out of the 629 CrebA-regulon genes were significantly downregulated in ovaries from HSD females (odds ratio = 3.75 and Fisher's exact $p = 7 \times 10^{-19}$; Fig. 3F). In addition, the mRNA reductions of three tested genes, *CrebA*, *Yp1*, and *Vm26Aa*, were readily verifiable in ovaries from HSD females (Fig. 3G). These results suggested an important role of the transcription factor CrebA in regulating vitellogenesis and HSD sensitivity.

## CrebA is required for yolk protein expression in follicle cells

To experimentally evaluate the function of CrebA in follicle cells, we used *tj-Gal4* to drive two RNAi lines to knock down *CrebA*. While both lines had a severe reduction in female fecundity, *tj>CrebA^RNAi-1* females completely failed to lay eggs (Figs.

4A and EV4B for RT-qPCR results showing the RNAi efficiencies). In contrast, *tj*-driven knockdown of *CrebB* had no effect on female fecundity. To identify developmental defects upon follicle cells-specific knockdown of *CrebA*, we quantified the distributions of egg chambers at different oogenesis stages in dissected ovaries. Here the fraction of stages-13–14 egg chambers exhibited a dramatic drop, from 14.6 ± 0.9% in *tj > +* (the control group) to 0% in *tj>CrebA^RNAi-1* and 11.0 ± 3.3% in *tj>CrebA^RNAi-2* (Fig. 4B pink). Notably, stage-9 egg chambers had a significant increase, from 10.2 ± 1.1% in the control group to 16.1 ± 0.3% in *tj>CrebA^RNAi-1* and 20.7 ± 0.1% in *tj>CrebA^RNAi-2* (Fig. 4B purple). Stage 9 coincides with the surges of both InR activity and *CrebA* expression in follicle cells. Consistent with a temporal role of CrebA, the *C204-Gal4* driver, which is active in follicle cells of stages 8 ~ 14, was also sufficient to lead both *CrebA^RNAi-1* and *CrebA^RNAi-2* to reduce the fraction of late-stage egg chambers (from 18.7 ± 1.2 to 1.7 ± 0.7 and 2.7 ± 1.2, respectively).

To identify the transcriptomic changes induced by *CrebA* knockdown, we performed bulk RNA-seq in *tj>CrebA^RNAi-1*, *tj>CrebA^RNAi-2* and *tj > +* ovaries. GO analysis revealed enrichment of downregulated genes in secretory functions, such as "membrane", "egg chorion", "extracellular region", "vitelline membrane formation involved in chorion-containing eggshell formation" (Fig. 4D). These pathways contain the major yolk protein genes and vitelline membrane genes, aligning with the enrichment of CrebA target genes. In particular, *CrebA*, *Yp1*, *Yp2*, *Vm26Aa*, and *Vm26Ab* all exhibited a significantly decreased expression in both RNAi lines but more prominently in *tj>CrebA^RNAi-1* (Figs. 4C and EV4B for RT-qPCR validations), consistent with the observed phenotypic severity of these two lines.

To further examine the protein expression of Yp1 during oogenesis, we used a *Yp1-GFP* line, which is a large genomic clone of *Yp1* gene with a green fluorescent protein tag (Sarov, Barz et al, 2016). In the ovaries of the control line, Yp1-GFP began to express in follicle cells and accumulate in the oocyte from stage 8, marking the onset of vitellogenesis (Fig. EV5A). As oogenesis progresses, Yp1-GFP in follicle cells peaked at stage 10a and became barely detectable at stages 13–14. Meanwhile, Yp1-GFP in the oocytes was continuously accumulated and granulized (Fig. EV5B). When *CrebA* was knocked down in follicle cells, Yp1-GFP became virtually undetectable in follicle cells throughout oogenesis, with weak signals in the oocytes only (Fig. 4E). In contrast, when *CrebA*

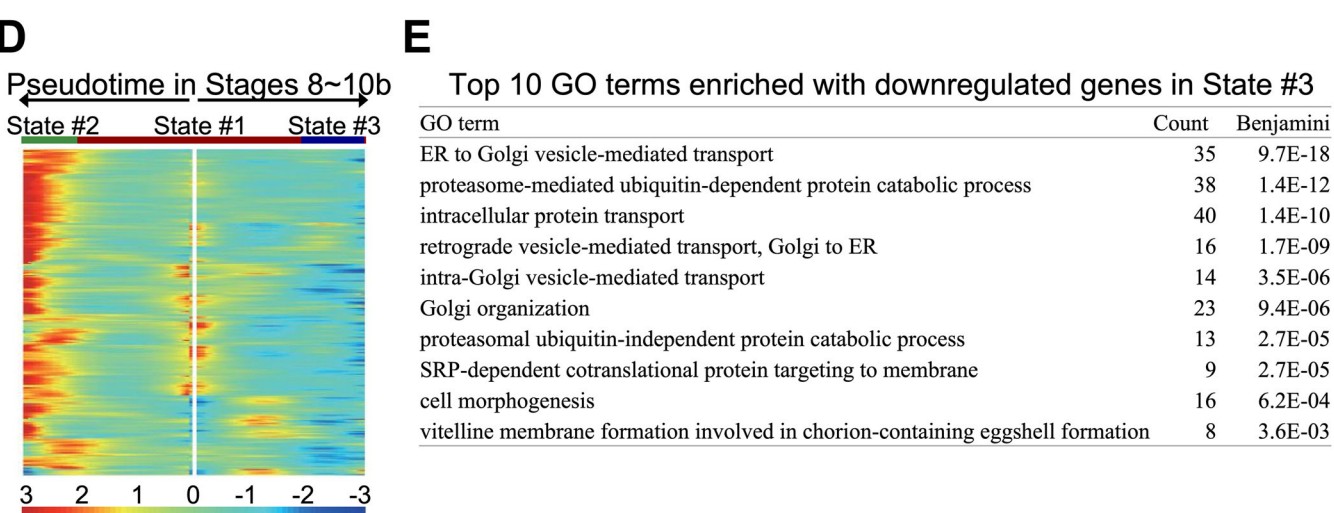

**Figure 2. HSD dampens the InR activity surge and leads to an abnormal subpopulation of stage-8 follicle cells.**

(A) A UMAP projection of Harmony-integrated ovarian scRNA-seq datasets prepared from *w1118* females under ND (red) or HSD (blue). (B) 18 single-cell clusters are annotated for cell types or stages according to marker genes in Fig. EV2. (C) 694 stages-8–10b follicle cells (ND, red; HSD, blue; stage 8, cyan; stages 9–10b, purple) ordered along pseudotime (black line). A new branch with 200 cells from HSD but none from ND was identified as State #3 (dark blue), as opposed to State #1 (dark red) and State #2 (dark green) that were shared between ND and HSD. (D) A pseudotime-ordered heatmap for significantly downregulated genes in State #3 (adjusted *p* < 0.05). (E) Top ten gene ontology terms enriched with genes from (D). Source data are available online for this figure.

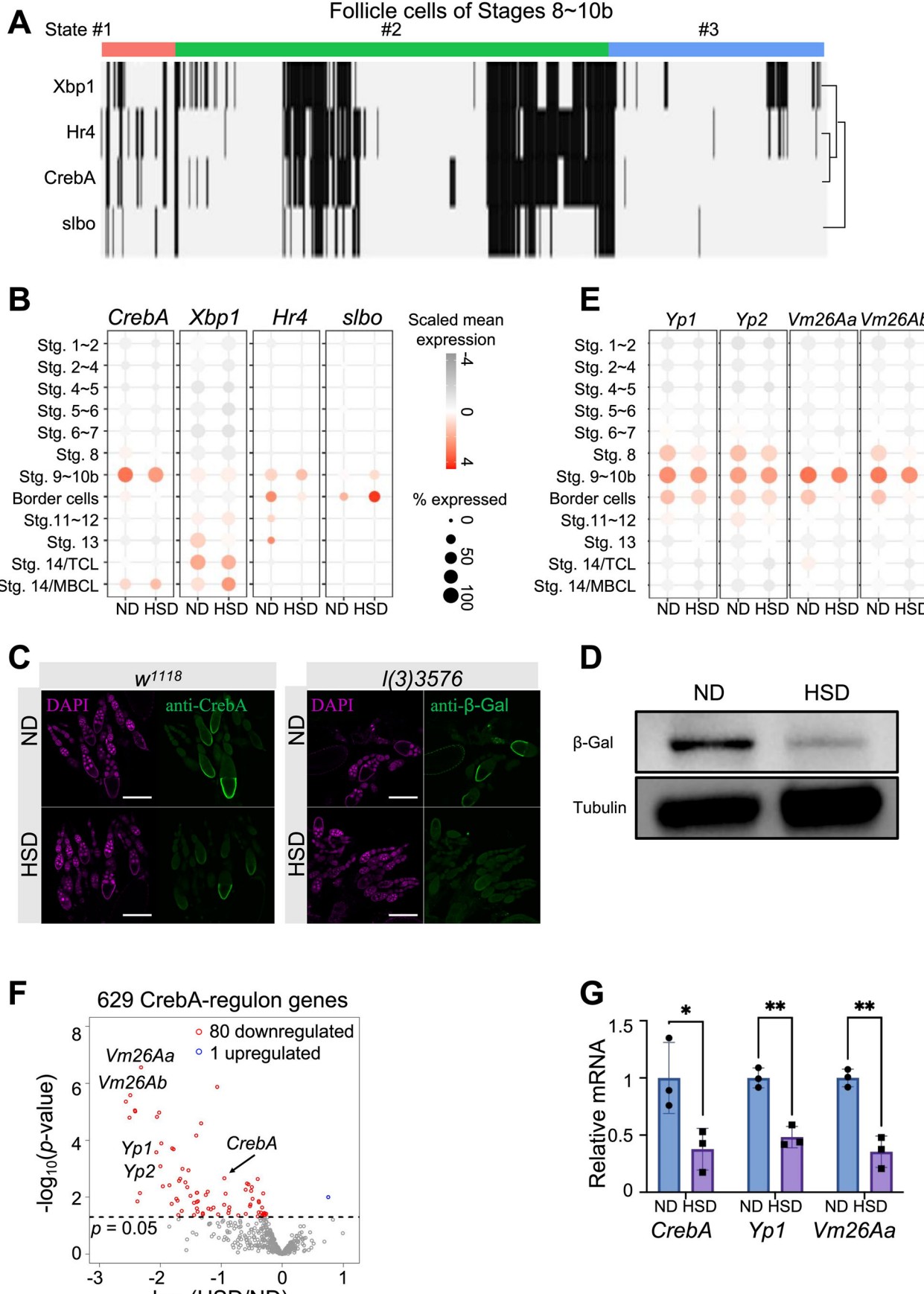

**Figure 3. The HSD-specific subpopulation of stage-8 follicle cells has a lower activity of CrebA-mediated transcriptional regulon.**

(A) Transcriptional regulons that were specifically switched off in follicle cells at State #3. Black and white bars represent cells in which the corresponding regulons were on or off (Fig. EV3 for all the active regulons in stages 8–10B follicle cells). (B) Single-follicle cell mRNA expression of the four HSD-deactivated transcription factors identified in (A). (C) Immunostaining with anti-CrebA (left) or anti-β-Gal (right) in egg chambers from females of *w1118* or *l(3)3576* (a lacZ reporter inserted in the first intron of *CrebA*), respectively. This reporter line expresses β-gal in the majority of the tissues that normally express CrebA, including follicle cells at vitellogenic stages, and meanwhile disrupts the expression of CrebA from its locus (Andrew et al, 1997; Rose et al, 1997). While the line might not fully report all aspects of CrebA in wild-type ovaries, the immunostaining patterns of CrebA and β-Gal resembled each other in the ovaries. Scale bars = 50 μm. (D) Western blot analysis with anti-β-Gal in ovaries from *l(3)3576* females under ND and HSD, respectively. The result was reproduced in three independent replicate experiments. The anti-CrebA antibody failed to work in Western blot experiments. (E) Single-follicle cell mRNA expression of four target genes in the CrebA-mediated regulon: *Yp1, Yp2, Vm26Aa,* and *Vm26Ab*. (F) Ovarian bulk RNA-seq confirms reduced expression of the 629 target genes (with DESeq2-computed Wald test *p* values <0.05) in the CrebA-mediated regulon by HSD, including *CrebA, Yp1, Yp2, Vm26Aa,* and *Vm26Ab*. (G) RT-qPCR confirms the reduced ovarian levels of *CrebA, Yp1,* and *Vm26Aa* mRNA by HSD. For each experimental group, seven pairs of ovaries were pooled as one sample and three biological samples were used. Error bars represent mean ± SEM. Student's *t*-test *p* = 0.04 (denoted as *), 0.002 (**), and 0.002 (**), respectively. Source data are available online for this figure.

was overexpressed in follicle cells, we observed an increase of Yp1-GFP in follicle cells during stages 8–9 and this expression further lingered until late-stage 10b (Fig. EV5C). These results support a role of CrebA in controlling the expression of yolk proteins in follicle cells.

In addition to its expression in follicle cells, CrebA is also known to be expressed in the fat body, another source of yolk protein synthesis (Abel, Bhatt et al, 1992; Bownes, Ronaldson et al, 1993; Sondergaard, Mauchline et al, 1995). We used *cg-Gal4* and *ppl-Gal4* to drive *CrebA*^RNAi specifically in the fat body, and detected no significant changes in either female fecundity or ovarian morphology (Appendix Fig. S2A,B). It is noted that *CrebA* mRNA level in the female fat body was unaffected by *tj>CrebA*^RNAi (Appendix Fig. S2C), suggesting a minimal fat body involvement in *tj-Gal4* action in our system (Weaver, Ma et al, 2020). In fact, both *Yp1* mRNA and Yp1-GFP levels in the fat body were actually increased in *tj>CrebA*^RNAi females (Appendix Fig. S2D). Together, these results suggest that the primary action site for CrebA in inducing Yp1 expression is the follicle cells.

## CrebA has a role in vitelline membrane formation

Vitelline membrane protein genes encode the major proteins that constitute the oocyte proximal layer of eggshell. A reduced expression of *Vm26Aa* and *Vm26Ab* could lead to a compromised structural integrity in this eggshell layer (Burke, Waring et al, 1987; Schupbach and Wieschaus, 1991). To directly evaluate this aspect of the defect that may have been caused by *CrebA* RNAi, we analyzed eggshell ultrastructure under transmission electron microscopy (TEM). Control egg chambers at stage 12 had a continuous vitelline membrane layer (marked as "vm" in Fig. 4F) with a thickness of 1.33 ± 0.11 μm when the chorionic layer ("ch") began to form, and the nuclei of main-body follicle cells ("fc") became flattened. At stage 14, while the vitelline membrane layer was thinned to 0.40 ± 0.06 μm and the chorionic layer was finalized, both the cytoplasm and the nuclei of follicle cells were flattened. In contrast, the most developed egg chambers of *tj>CrebA*^RNAi-1 and *tj>CrebA*^RNAi-2 females exhibited a much thinner layer of vitelline membrane (0.15 ± 0.05 and 0.21 ± 0.03 μm, respectively) and a substantial layer of chorion without specified structures such as periodic cavities ("cv" in Fig. 4F). In addition, the very few eggs deposited by *tj > CrebA*^RNAi-2 females exhibited morphologically deformed eggshells and failed to develop (Appendix Fig. S3). These

results documented a vital role of CrebA in follicle cells in controlling vitelline membrane formation.

## Overexpression of CrebA in follicle cells rescues reproductive phenotypes caused by HSD or follicle-cell expression of dominant negative InR

To test whether CrebA is crucial for female reproductive physiology in a manner sensitive to HSD or disrupted insulin signaling activity in follicle cells, we used *tj*-GAL4 to overexpress *UAS-CrebA* (*tj>CrebA*) in *tj > +* females on HSD or in *tj > InR*^DN females. Similar to *tj > CrebA*^RNAi-1 and *tj>CrebA*^RNAi-2 females (Fig. 4A), both HSD females and *tj > InR*^DN females were subjected to reduced fecundity (Fig. 5A,B purple). Importantly, their reproductive outcomes were largely restored by *tj>CrebA* (Fig. 5A,B green). Thus, follicle-cell-expressed CrebA can counteract the effects of HSD and InR^DN on reproductive physiology. In addition, we successfully verified the rescue effects of *tj>CrebA* for the following three aspects: ovarian *Yp1* and *Vm26Aa* mRNAs (Fig. 5C,D), Yp1-GFP expression in follicle cells (Fig. 5E,F), and eggshell ultrastructure (Fig. 5G–I). Together, these results support CrebA as a downstream effector of dampened InR activity in follicle cells during vitellogenesis.

To evaluate the rescue effect of *tj>CrebA* at the transcriptomic level, we performed ovarian RNA-seq in *tj > +* females under ND, *tj > CrebA* females under ND, *tj > +* females under HSD, and *tj > CrebA* females under HSD. Figure 5C shows a negative whole-transcriptome correlation between the dietary effect (i.e., contrast between HSD and ND) and the rescue effect (i.e., contrast between *tj > CrebA* and *tj > +* under HSD). For example, *CrebA, Yp1, Yp2, Vm26Aa,* and *Vm26Ab* mRNAs were all significantly reduced by HSD, but *tj>CrebA* counteracted their reductions (Fig. 5C colored markers). Notably, six outlier genes were dramatically upregulated by HSD (Fig. 5C squares). Five of them, *Jon65Aiii, Jon99Fii, thetaTry, SP151,* and *yip7*, encode serine peptidases (SPs) that are highly expressed in the guts but not in the ovaries. Thus, HSD likely exerts two distinct types of impact on the secretory capacity of the ovary: to decrease the synthesis of yolk and vitelline membrane proteins by deactivating CrebA, and to ectopically activate SPs independent of CrebA. This CrebA-independent effect of HSD may be related to our observed partial rescue of HSD female fecundity by CrebA as opposed to the better rescue for *tj > InR*^DN females (Fig. 5A,B), a possibility that requires future investigations.

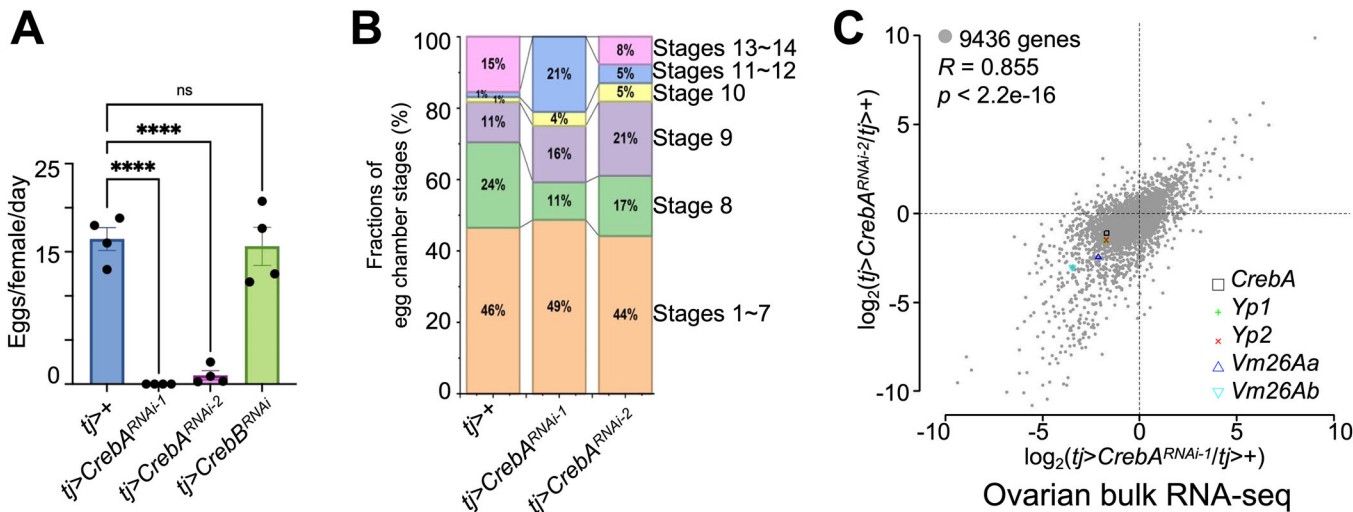

D

## Top 10 GO enriched with genes downregulated by *tj>CrebA^RNAi*

| GO terms | Count | Benjamini |
|---|---|---|
| transmembrane transport | 56 | 1.6E-13 |
| membrane | 174 | 4.3E-12 |
| egg chorion | 20 | 1.7E-11 |
| plasma membrane | 122 | 3.8E-10 |
| transmembrane transporter activity | 33 | 3.3E-07 |
| extracellular region | 44 | 4.1E-06 |
| mesoderm development | 15 | 2.5E-04 |
| extracellular matrix structural constituent | 10 | 2.0E-04 |
| vitelline membrane formation involved in chorion-containing eggshell formation | 8 | 4.9E-04 |
| muscle contraction | 7 | 4.9E-04 |

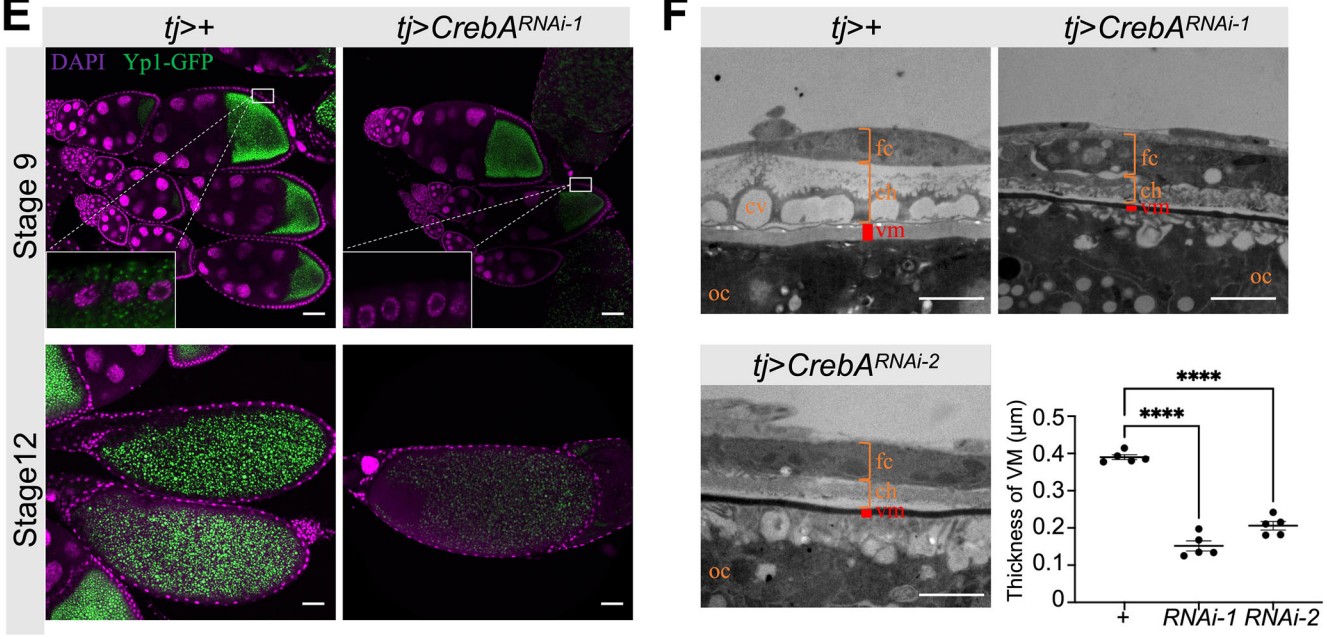

Figure 4.   Follicle cell-specific expression of CrebA is required for successful folliculogenesis.

(A) Fecundity of *tj* > +, *tj>CrebA*<sup>RNAi-1</sup>, *tj>CrebA*<sup>RNAi-2</sup>, and *tj>CrebB*<sup>RNAi</sup> females on days 4–6 after eclosion. N = 4 biological replicates for each genotype. Error bars represent mean ± SEM. One-way ANOVA p < 0.0001 (denoted as ****), < 0.0001 (****), and =0.73 (denoted as ns) from left to right, respectively. (B) Distributions of egg chambers at different stages from *tj* > +, *tj>CrebA*<sup>RNAi-1</sup>, and *tj>CrebA*<sup>RNAi-2</sup> females. N = 6, 4, and 5 females, respectively. (C) A scatter plot showing good correlation of transcriptomic changes between *tj>CrebA*<sup>RNAi-1</sup> and *tj>CrebA*<sup>RNAi-2</sup> ovaries. Pearson's correlation was computed based on 9436 genes that were expressed in all the samples: correlation coefficient R = 0.855 and p value is approaching 0. *CrebA*, *Yp1*, *Yp2*, *Vm26Aa*, and *Vm26Ab* were highlighted. (D) Top 10 GO enriched with genes downregulated by *tj>CrebA*<sup>RNAi-1</sup>. (E) Representative images showing reduced levels of Yp1-GFP (green; DAPI, magenta) in follicle cells at stage 9 and oocytes at stage 12 from *tj* > *Yp1-GFP*; *CrebA*<sup>RNAi-1</sup> ovaries. Scale bars = 50 μm. (F) Representative transmission electron microscope (TEM) images showing aberrance in eggshell structure of the most developed egg chambers from *tj>CrebA*<sup>RNAi-1</sup> and *tj>CrebA*<sup>RNAi-2</sup> females. Scale bars = 2 μm. The major layers from the exterior of the egg to the interior are follicle cells (fc), chorionic layers (ch), vitelline membrane (vm), and the oocyte (oc). We note two common phenotypes: (1) missing periodic cavities (cv), and (2) thinner vitelline membrane (vm). For quantification of vm thickness, each genotype was measured in N = 5 samples prepared from different egg chambers. Error bars represent mean ± SEM. One-way ANOVA p values <0.0001 (denoted as ****) for both RNAi strains. Source data are available online for this figure.

## *CrebA* is transcriptionally repressed by FoxO in the ovaries

In response to IIS attenuation, FOXO transcription repressors are translocated to the nucleus and inhibit cellular growth (Biggs, Meisenhelder et al, 1999; Brunet, Bonni et al, 1999; Hwangbo, Gersham et al, 2004; Wang, Moya et al, 2011). To test whether *CrebA* is a direct target gene of FoxO, we analyzed the publicly available ChIP-seq data (Kudron et al, 2018; Data ref: Kudron et al, 2018) and identified two FoxO binding peaks at the *CrebA* locus, each of which carries one site matching the consensus motif (Fig. 6A,B). To validate the FoxO binding at these two sites in follicle cells, we performed ChIP-qPCR using anti-GFP in *tj>FoxO-GFP* ovaries. Figure 6C shows that, while there is a weak enrichment at these sites (fold difference relative to IgG = 2.0 ± 1.4 and 2.1 ± 0.8, respectively), such binding was greatly enhanced in the presence of *InR*<sup>DN</sup> (fold difference = 14.7 ± 6.8 and 10.3 ± 6.0, respectively; Fig. 6C light bars). Consistently, the ovarian *CrebA* mRNA was reduced in *tj>FoxO-GFP*;*InR*<sup>DN</sup> females but not in *tj>FoxO-GFP* females (Fig. 6D). These results suggest an inhibitory role of InR activity in preventing the FoxO repressor from inhibiting *CrebA* gene transcription.

To evaluate the inhibitory effect of FoxO on CrebA expression in follicle cells, we co-stained the ovaries with anti-GFP and anti-CrebA. In *tj* > *FoxO-GFP* ovaries, the nuclear-to-cytoplasmic ratio of FoxO-GFP in follicle cells was significantly reduced during stages 9 ~ 10 and exhibited a recovery at stage 11 (Fig. 6E,F green), coinciding with a dynamic change in CrebA signals (Fig. 6E,F red). In contrast, in *tj>FoxO-GFP*;*InR*<sup>DN</sup> ovaries, FoxO-GFP was precociously translocated into the nuclei at stage 9, accompanied by a significant reduction of CrebA expression (Fig. 6G). These results further support a model in which FoxO-mediated inactivation of CrebA expression becomes relieved by an InR activity surge during early vitellogenesis.

## FOXO1-CREB3L2 regulation represents the human counterpart of FoxO-CrebA

While CrebA is the sole *Drosophila* member of CREB3-like bZIP transcription factors, the human genome has five genes, *CREB3* and *CREB3L1-4* (Khan and Margulies, 2019). Among them, CREB3L2 encodes a protein that shares the best homology with CrebA in the bZIP DNA binding domain (Fig. 7A,B) and in the CREB3s-unique domain (Barbosa, Fasanella et al, 2013). To test whether these genes may exert CrebA-like functions in human ovarian follicle cells, we

treated KGN cells, a granulosa cell line, with *FOXO1* siRNAs. Whereas *CREB3L3* and *CREB3L4* had no mRNA expression in these cells, the other three genes were significantly increased in their expression level when *FOXO1* was knocked down (Fig. 7C), suggesting transcriptional suppression of CREB3-like proteins by FOXO1. To further evaluate the functional similarity between human CREB3-like proteins and CrebA, we knocked down *CREB3L2* in KGN cells (Fig. 7E) and performed RNA-seq. We found that the 128 significantly downregulated genes were enriched in GO and KEGG terms of "extracellular region", "female pregnancy", "hormone activity", "Cytokine-cytokine receptor interaction", "PI3K-Akt signaling pathway", and "JAK-STAT signaling pathway" (Fig. 7D). This result aligns with the role of CrebA in regulating secretory capacity and insulin signaling in *Drosophila* follicle cells. Importantly, the expression level of *CYP19A1*, which encodes the aromatase that converts androgens to estrogens, was reduced by ~42.2-fold under *CREB3L2* siRNA treatment. Western blot analysis confirmed this reduction (Fig. 7F). Consistently, 17β-estradiol (E2), the major circulating estrogen and a product of CYP19A1, was significantly reduced by *CREB3L2* siRNAs (Fig. 7G). Importantly, knockdown of *FOXO1* effectively counteracted the effects of *CREB3L2* siRNAs on *CREB3L2* mRNA, CYP19A1 protein and E2 levels (Fig. 7H–J). Together, our results suggest that human FOXO1-CREB3L2 regulation, similar to FoxO-CrebA in *Drosophila*, also controls the secretory capacity of ovarian follicle cells in a highly conserved manner.

# Discussion

The precise coordination of developmental and metabolic cues is essential for successful reproduction. Using a phase separation-based reporter, we detect a surge of InR kinase activity specifically in vitellogenic follicle cells, which coincides with the heightened biosynthetic demands at this developmental time. This stage-specific surge serves as a critical driver for folliculogenesis, fueling biosynthesis of yolk and vitelline membrane proteins through the transcription factor CrebA. These findings not only resolve long-standing questions about the stage-specific roles of IIS in somatic follicle cells but also shed light on how external signals and intracellular regulatory networks coordinate to gate developmental transitions, with broader implications for understanding metabolic dysregulation in human conditions such as PCOS, where follicular arrests partially mirror the IIS-CrebA axis disruption that we discuss in this report.

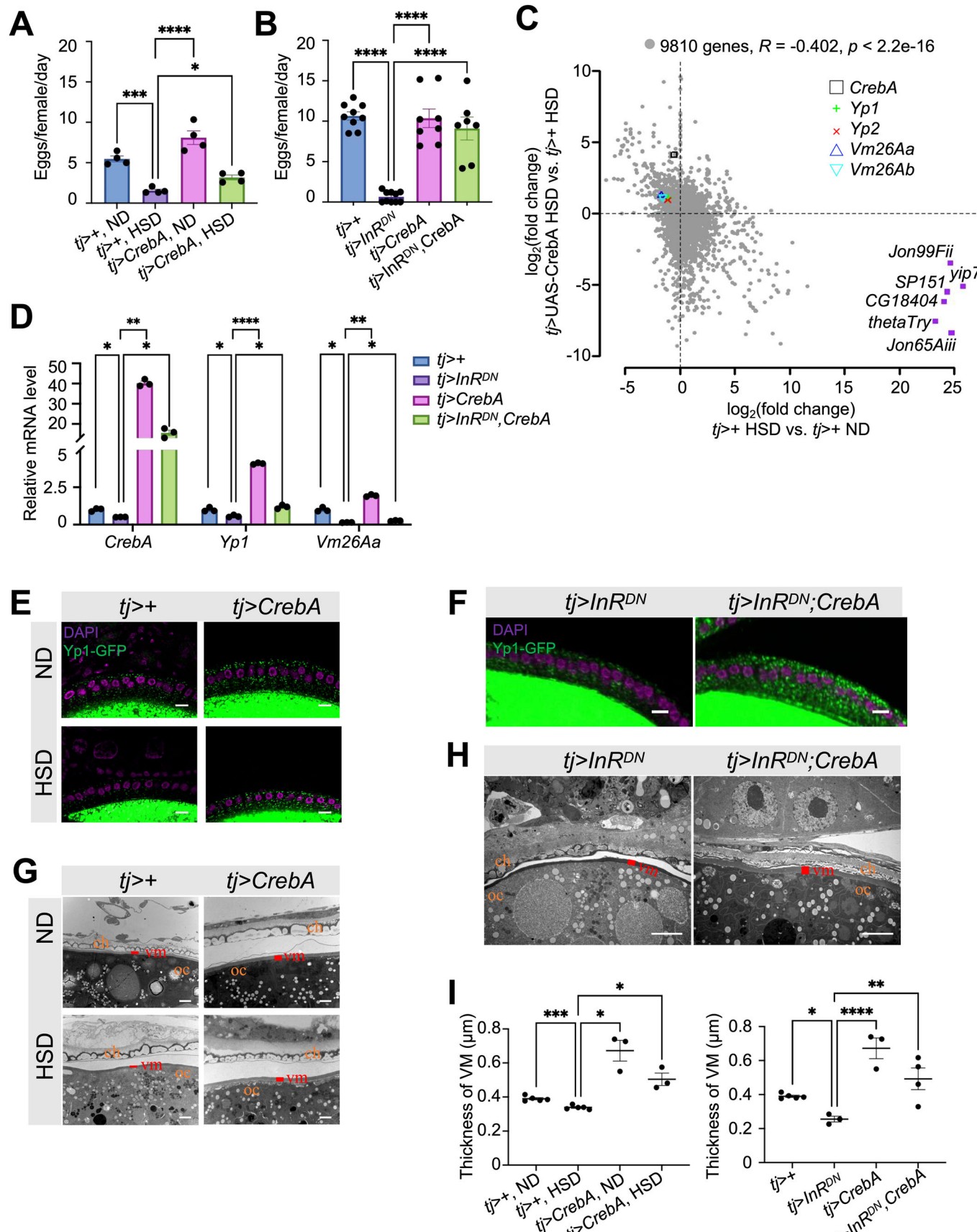

**Figure 5. Follicle cell-specific overexpression of CrebA rescues the subfertility induced by HSD or dominant negative InR.**

(A) Female fecundity measured for *tj* > + under ND, *tj* > + under HSD, *tj>CrebA* under ND and *tj* > *CrebA* under HSD on days 7–9 after eclosion. N = 4 biological replicates for each group. Error bars represent mean ± SEM. One-way ANOVA *p* values = 0.0001 (denoted as ***), <0.0001 (****) and =0.04 (*) for the marks from left to right, respectively. (B) Fecundity measured for *tj* > +, *tj* > *InR^DN^*, *tj* > *CrebA* and *tj* > *InR^DN^;CrebA* females on days 4–6 after eclosion. N = 9, 10, 8, and 6 biological replicates, respectively. Error bars represent mean ± SEM. One-way ANOVA, all *p* values <0.0001 (denoted as ****). (C) A scatter plot shows a negative correlation between the HSD-to-ND fold changes and the rescue-to-control (*tj>CrebA* vs. *tj* > +) fold changes in ovarian bulk RNA-seq. Pearson's correlation was computed based on 9810 genes that were expressed in all the samples: R = −0.402, and *p* value is approaching 0. *CrebA, Yp1, Yp2, Vm26Aa, Vm26Ab*, and 6 outlier genes were highlighted. (D) RT-qPCR measurements of *CrebA, Yp1*, and *Vm26Aa* mRNAs in ovaries from *tj* > +, *tj* > *InR^DN^*, *tj>CrebA* and *tj* > *InR^DN^;CrebA* females. *Yp1-GFP* was also present in these flies. N = 3 biological replicates. Error bars represent mean ± SEM. One-way ANOVA *p* = 0.03 (*), 0.001 (**) and 0.02 (*) for *CrebA; p* = 0.04 (*), <0.0001 (****), and 0.01 (*) for *Yp1; p* = 0.02 (*), 0.001 (**), and 0.04 (*) for *Vm26Aa*. (E, F) Representative images showing that the levels of Yp1-GFP (green; DAPI, magenta) in stage-9 follicle cells and oocytes are reduced by HSD (E) or by *tj* > *InR^DN^* (F) and can be recovered by *tj>CrebA*. Scale bars = 10 μm. (G, H) Representative TEM images showing that the vitelline membrane thickness of stage-14 egg chambers is reduced by HSD (G) or *tj* > *InR^DN^* (H) and can be recovered by *tj>CrebA*. The chorionic layer, the vitelline membrane and the oocyte are labeled as "ch", "vm", and "oc". Scale bars = 2 μm. (I) Quantification of vitelline membrane thickness from TEM experiments in panels G and H. N = 5, 5, 3, 3, 3, and 4 biological replicates from left to right, respectively. Error bars represent mean ± SEM. One-way ANOVA *p* values = 0.0004 (***), 0.03 (*), 0.04 (*), 0.05 (*), <0.0001 (****), and 0.003 (**) for the significance marks, respectively. Source data are available online for this figure.

Our previous study suggested an involvement of IR in HSD-induced oogenesis defects (Liu et al, 2022), and our current work further advances our knowledge. Here, we document that the IIS-mediated gene regulatory network has a direct role in vitellogenesis and egg maturation. Reduction of InR activity in follicle cells triggers nuclear import of FoxO and inhibition of CrebA expression (Fig. 6G). The stage-specific sensitivity of follicle cells to IIS disruption highlights vitellogenesis as a metabolic checkpoint. Importantly, this study suggests that CrebA acts as an intracellular executor of this checkpoint. Our results show that CrebA overexpression can rescue the oogenesis phenotypes caused by HSD and InR^DN, suggesting that enhancing the secretory capacity of follicle cells could mitigate diet-induced infertility. Whether this strategy is applicable to vertebrates requires future exploration. Nevertheless, the functional conservation between CrebA and CREB3L2 underscores the importance of secretory pathway regulation in follicle cell function. Future studies exploring CREB3-like proteins in clinical cohorts may open therapeutic avenues for metabolic reproductive disorders.

CrebA is a CREB/ATF-family transcription factor best known for its master regulator role of secretory machinery in embryonic and larval tissues (Abrams and Andrew, 2005; Fox, Hanlon et al, 2010; Jackson, Peng et al, 2025; Johnson, Wells et al, 2020). In the epidermis, tracheae, salivary glands, and sensory neurons, CrebA activates genes encoding components of the endoplasmic reticulum (ER) and Golgi apparatus, including COPI/COPII vesicle coat proteins and glycosylation enzymes, thereby amplifying the secretory capacity of these cells (Bhuiyan, Bordet et al, 2023; Fox et al, 2010; Iyer, Iyer et al, 2013; Johnson et al, 2020). In the fat body, CrebA is reported to have a role in the synthesis of antimicrobial, hemolymph, and yolk proteins (Abel et al, 1992; Troha, Im et al, 2018). However, its involvement in adult ovarian follicle cells, which also have high secretory demands, remains largely unexplored. Our current study positions CrebA as a key effector of IIS in follicle cells, bridging nutrient sensing to transcriptional activation of secretory proteins required for vitellogenesis and oocyte maturation. It is worth noting that there appears to be a compensatory upregulation of follicle cell-expressed Yp1 induced by fat-body *CrebA* knockdown (Appendix Fig. S2), suggesting that CrebA may have a role in mediating the interaction between systemic and ovarian IIS.

Our study establishes that FoxO-mediated repression of CrebA in follicle cells is subject to negative regulation by InR activity. It

remains to be resolved whether this negative regulation is solely responsible for the stage-specific wave of *CrebA* transcription or whether there is also an involvement of stage-specific transcriptional activators for driving *CrebA* expression. Nonetheless, we treated cultured ovaries with the addition of recombinant human insulin, and found that *CrebA* mRNA level was significantly increased (Appendix Fig. S4). This result supports the importance of IIS in transcriptional up-regulation of *CrebA* in the ovary. It is noteworthy that, in addition to its expression in follicle cells during vitellogenic stages 8–10b, *CrebA* is also expressed at the maturation stage 14 (Fig. 3C), a temporal pattern that coincides with the two waves of ecdysteroid signaling activity (Hackney, Pucci et al, 2007; Knapp and Sun, 2017; Sun, Smith et al, 2008). Future work is required to map out systematically the precise interplay between diet, IIS, ecdysone, and secretory regulation in achieving a delicate balance between metabolism and reproduction.

Our observed InR activity surge in vitellogenic follicle cells raises an interesting question about its origin, given that all egg chambers are expected to be accessible to circulating ILPs regardless of stage. One possibility is a stage-specific burst of InR expression, but, based on our scRNA-seq data, we found no evidence of such an expression increase in vitellogenic follicle cells for either InR or its binding partner Chico. Alternatively, autocrine signaling of insulin-like peptides (ILPs) might have a role, but our scRNA seq data did not detect any significant expression of *ILP1-5* in follicle cells, and *ILP6-8* expression was detected predominantly only during post-vitellogenic stages. Interestingly, we found that *pico*, which encodes a Ras-responsive intracellular adapter protein, is highly expressed in follicle cells at early and late stages with a marked reduction during mid-oogenesis. Given that its mammalian homologs, GRB10/14, can inhibit InR activity via direct binding (Holt and Siddle, 2005; Kim, Semple et al, 2015), it is tempting to speculate that the vitellogenic InR surge may be related to a stage-specific relief of Pico-mediated repression of InR activity. In this context, it is worth noting that, in our SPARK analysis, a constitutively active form of InR failed to raise SPARK levels either prior to or post the surge. Future investigations are required to elucidate the mechanisms responsible for the InR activity surge in vitellogenic follicle cells. Regardless of the precise mechanistic details, it is important to emphasize that this InR activity surge is subject to perturbation through compromising IIS in follicle cells (Fig. 1) and that our HSD-mediated IR model has provided a valuable tool toward the delineation of a conserved

regulatory mechanism important for folliculogenesis and egg maturation.

# Methods

### Reagents and tools table

| Reagent/resource | Reference or source | Identifier or catalog number |
| --- | --- | --- |
| **Experimental models** | | |
| *w1118* | Bloomington Drosophila Stock Center | 3605 |
| *ppl-Gal4* | Bloomington Drosophila Stock Center | 58768 |
| *C204-Gal4* | Bloomington Drosophila Stock Center | 50286 |
| *CrebA-lacZ* | Bloomington Drosophila Stock Center | 10183 |
| *UAS-CrebA* | Bloomington Drosophila Stock Center | 79021 |
| *UAS-FoxO-GFP* | Bloomington Drosophila Stock Center | 43633 |
| *UAS-InR$^{DN}$* | Bloomington Drosophila Stock Center | 8253 |
| *UAS-CrebA$^{RNAi-1}$* | TsingHua Fly Center | THU2277 |
| *UAS-CrebA$^{RNAi-2}$* | TsingHua Fly Center | THU2828 |
| *UAS-CrebB$^{RNAi}$* | TsingHua Fly Center | THU2514 |
| *Yp1-GFP* | Vienna Drosophila Resource Center | 318746 |
| *tj-Gal4* | Xiaohang Yang at Zhejiang University | N/A |
| *cg-Gal4* | Xiaohang Yang at Zhejiang University | N/A |
| *UAS-InR$^{CA}$* | Yan Yan at Hong Kong University of Science and Technology | N/A |
| *UAS-InR$^{SPARK}$* | Hai Huang at Zhejiang University | N/A |
| KGN cells (*H. sapiens*) | Procell | CL-0603 |
| **Antibodies** | | |
| CrebA Rbt-PC | Developmental Studies Hybridoma Bank | AB_10805295 |
| Anti-β-Galactosidase | Promega | Z3781 |
| chk-anti-GFP | Abcam | ab13970 |
| CYP19A1 Polyclonal antibody | Proteintech | 16554-1-AP |
| HRP-conjugated β-Tubulin Mouse mAb | Abclonal | AC030 |
| Cy3 affinipure Goat anti-Rabbit IgG (H + L) | Jackson immunoresearch | 111-165-144 |
| Goat anti-Mouse IgG (H + L) Cross-Adsorbed Secondary Antibody, Cyanine3 | Thermo Fisher Scientific | A10521 |

| Reagent/resource | Reference or source | Identifier or catalog number |
| --- | --- | --- |
| Goat anti-Chicken IgY (H + L) Secondary Antibody, Alexa Fluor™ 488 | Thermo Fisher Scientific | A-11039 |
| **Oligonucleotides and other sequence-based reagents** | | |
| **RT-qPCR primers** | | |
| **Gene** | **5'-3'** | |
| *Rp49* | F: GCTAAGCTGTCGCACAAATG R: GTTCGATCCGTAACCGATGT | N/A |
| *CrebA* | F: GACGGAGCACTCCTACAGTCT R: GAAATGGCGGGAAAGCACTC | N/A |
| *Yp1* | F: GTCTGGAGAACATGAACCTGGA R: GAGCAACGGTCTTGTCACCAT | N/A |
| *Vm26Aa* | F: TGACCCGTCTCCGTAAGTCT R: CAGGTAGTTCTTGGGGCAGG | N/A |
| *FoxO* | F: CCGCCAGCTTGGAAGATAATA R: CACGGGAAAGTTCTCCAGATT | N/A |
| *Actin* | F: CGACAGGATGCAGAAGGAG R: TCCTGCTTGCTGATCCACAT | N/A |
| *FOXO1* | F: TCGTCATAATCTGTCCCTACACA R: CGGCTTCGGCTCTTAGCAAA | N/A |
| *CREB3* | F: ATGCTGGTGACCAAGACCTG R: AGTCGCTCGGTACCTCAGAA | N/A |
| *CREB3L1* | F: GGAGAATGCCAACAGGACC R: GCACCAGAACAAAGCACAAG | N/A |
| *CREB3L2* | F: CACTGGGGTTGATTCCTCGTG R: AATGCAGGTGGTCCACTGGG | N/A |
| **ChIP-qPCR primers** | | |
| **Gene** | **5'-3'** | |
| *CrebA* | F: AGTGCGAACAAAACGCTCTTC R: TCATTGGTGCGCCCTTCTTT | N/A |
| *CrebA* | F: CTCCGATCTCCGGTCTCAAA R: AGTATCCGACTCTCTCCCGA | N/A |
| **siRNA** | | |
| **Gene** | **5'-3'** | |
| *FOXO1* | CCCAGUCUGUCUGAGAUAATT | N/A |
| *FOXO1* | CAAUUCGUCAUAAUCUGUCCCUA | N/A |
| *CREB3L2* | GAGUCUUGUUCAACUGAGATT | N/A |
| *CREB3L2* | ACCAAAUUGCCCCUGUCAATT | N/A |
| *GFP* | CAAGCUGACCCUGAAGUUCTT | N/A |
| **Chemicals, enzymes and other reagents** | | |
| Schneider's Insect Medium | Gibco | 21720024 |
| HBSS | Gibco | 1025076 |
| TrypLe | Gibco | 12605036 |
| RNAiso Plus | Takara | D9108A |
| collagenase | Sigma | C9722 |
| Human insulin | DulyBiotech | P0029 |

| Reagent/resource | Reference or source | Identifier or catalog number |
|---|---|---|
| ChromoTek GFP-Trap Magnetic Agarose beads | Proteintech | Gtma-20 |
| ABScript II RT Master Mix | ABclonal | RK20429 |
| SYBR Green Fast qPCR Mix | ABclonal | RK21203 |
| DMEM/F12 medium | Servicebio | G4610 |
| jetPRIME Versatile DNA/siRNA transfection reagent | Polyplus | 101000046 |
| FBS | GeminiBio | 900-108 |
| DAPI | Beyotime | P0131 |
| penicillin/ streptomycin | Biosharp | BL505A |
| **Software** | | |
| Fiji | FIJI | http://fiji.sc |
| GraphPad Prism 9 | GraphPad | https://www.graphpad.com |
| Origin 2025b | OriginLab | https://www.originlab.com |
| fastp v0.20.1 | Chen et al, 2018 | |
| HISAT2 v2.2.1 | Kim et al, 2019 | |
| featureCounts v2.0.1 | Liao et al, 2014 | |
| Harmony v1 | Qiu, Mao et al, 2017 | |
| Monocle v2 | Korsunsky, Millard et al, 2019 | |
| SCENIC v0.9.1 | Aibar et al, 2017 | |
| **Other** | | |
| FastPure Cell/Tissue DNA Isolation Mini Kit | Vazyme | DC102-01 |
| E2 ELISA Kit | ABclonal | RK00651 |
| ChIP-IT Express Enzymatic | Active Motif | 53009 |
| NovaSeq6000 | Illumina | |

## Fly husbandry

All flies were maintained at 25 °C and 60% relative humidity with an altered light/dark cycle. Three different food formulas were used. (1) The normal diet (ND) contained 8.6 g/dL corn flour, 1.2 g/dL agar, 3 g/dL dry yeast, and 2 g/dL sucrose. (2) The high-sucrose diet (HSD) increased the sucrose concentration to 35 g/dL. (3) Whenever unspecified, flies were maintained on a regular diet (RD), which contains 5 g/dL corn flour, 1 g/dL agar, 2.45 g/dL dry yeast, 0.725 g/dL sucrose, and 3 g/dL brown sugar. To measure fecundity and oogenesis stage distribution, we collected newly emerged adult females and allowed them to mate with w1118 males on RD for three days. Then females were isolated and transferred into new vials, each with 2–6 females on the specified diet. On a daily basis, the food was changed with the same formula and the number of eggs was counted. For comparisons between HSD and ND, the measurements were performed on days 4–6 after transferring because this duration of HSD treatment showed the strongest insulin resistance (Liu et al, 2022); for the other experimental settings, the measurements were performed on days 1–3 after transferring.

## Dissection of ovaries and immunohistochemistry

Ovaries were dissected in cold Schneider's Insect Medium (Gibco) with 10% FBS (GeminiBio). For the insulin treatment, freshly dissected ovaries were incubated in the medium with 0.5 μM recombinant human insulin (DulyBiotech) for 4 h before further experiments. For immunohistochemistry, freshly dissected ovaries were fixed in PBS with 4% formaldehyde. After washing, the samples were permeabilized in PBS with 1% Triton X-100, blocked in PBST with 3% BSA, incubated with the primary antibody at 4 °C overnight, and then incubated with the secondary antibody at RT for 1 h. After washing, the samples were mounted in Vectashield with DAPI (Beyotime).

## Fluorescence microscopy and image analysis

Fluorescence imaging was performed with an FV1000 confocal microscope. ImageJ/Fiji software was used for image processing and analyses. For quantitative analysis of InR$^{SPARK}$ in single-follicle cells, the cellular region and individual GFP spots in the cytoplasm were identified by the Analyze Particle function of Fiji. Then the summed pixel intensities of GFP spots divided by the total pixel intensities of a given cellular region was defined as the InR$^{SPARK}$ activity (Li et al, 2022). For each egg chamber, its developmental stage was classified according to (Jia, Xu et al, 2016).

## Transmission electron microscopy

Samples for TEM were prepared as previously described (Row, Huang et al, 2021). Sample sections were put into an H-7650 TEM (Hitachi) for observing at 80 kV of accelerating voltage. Interested regions of the membrane alteration were photographed by a Gatan 830 CCD camera (Gatan). The thickness of the vitelline membrane was quantified with Fiji.

## Bulk RNA sequencing and analysis

Total RNA were extracted with RNAiso Plus (Takara). cDNA libraries were prepared with VAHTS V3 Library Prep Kit (Vazyme) and sequenced with NovaSeq6000 (Illumina). Clean reads were trimmed for adapters with fastp v0.20.1 (Chen et al, 2018), and then aligned to the reference build dmel_r6.34 or hg38 with HISAT2 v2.2.1 (Kim et al, 2019). Read counts mapped to protein-coding genes were summarized with featureCounts v2.0.1 (Liao et al, 2014). Differentially expressed gene analysis was performed with DESeq2 v1.32.0 (Love, Huber et al, 2014). Functional enrichment analyses of differentially expressed genes ($|log2Fold-Change| > 1$ and $p$ value $<0.05$) were performed with https://davidbioinformatics.nih.gov/.

## Ovarian single-cell RNA sequencing

Freshly dissected ovaries were washed twice with iced PBS and then incubated with 300 μL mixed enzyme solution, collagenase (Sigma)

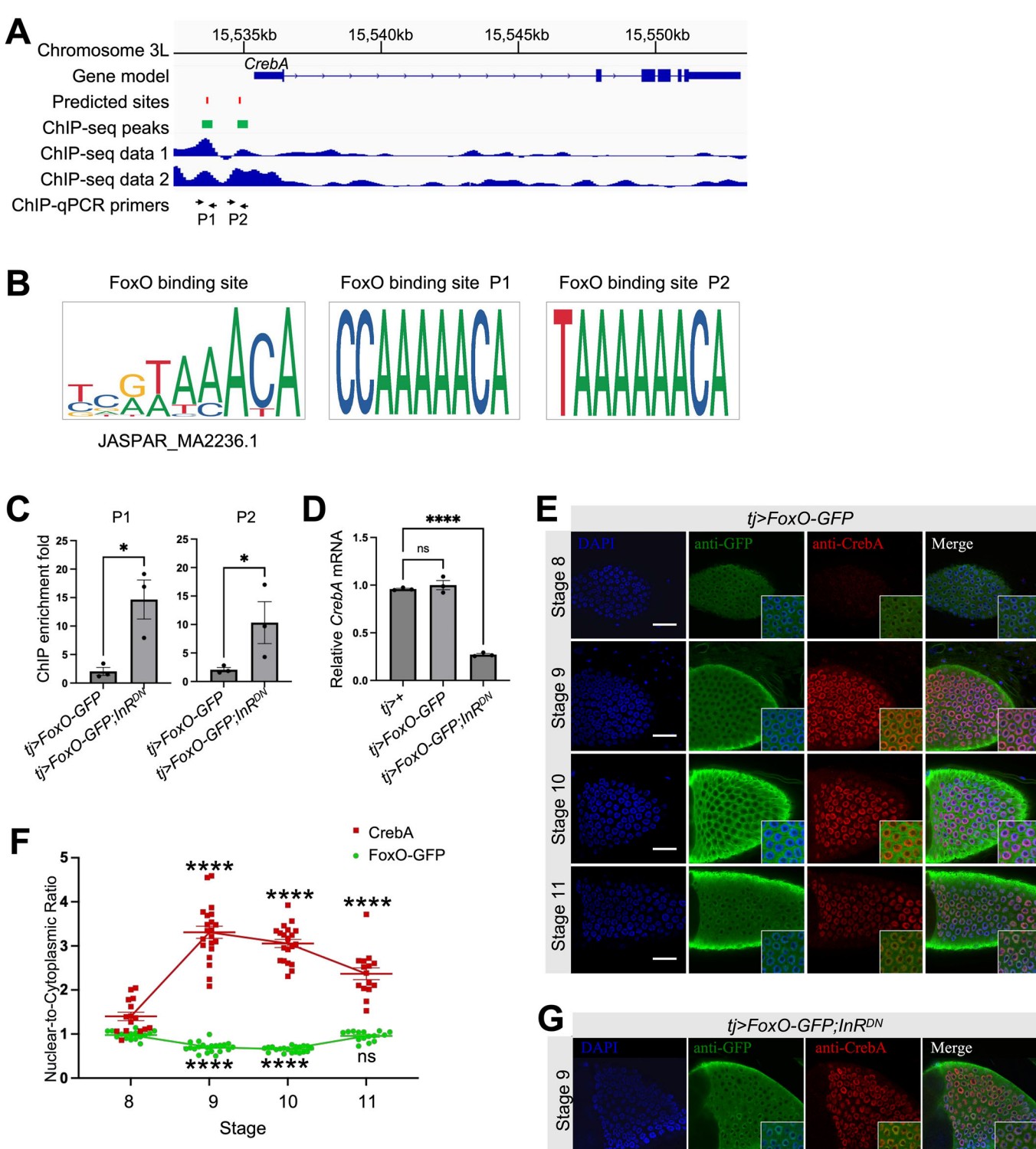

with TrypLe (Gibco). Tissue dissociation was facilitated by pipetting with a P200 every 5 min. After incubation at room temperature for 40 min, the solution was passed through a 60-μm nylon filter. After an additional 15 min of incubation, the reaction was stopped by adding 550 μL Schneider's Insect Medium with 10% FBS. The solution was then passed through another 30-μm nylon filter and centrifuged at 1200 rpm for 7 min. The cells were resuspended with precooled HBSS (Gibco). Cell suspensions were analyzed by quantification of AO&PI staining with Fluorescence Cell Analyzer (Countstar). For the ND and HSD samples, the concentrations were 653 cells/μL and 883 cells/μL, respectively; the viabilities were 90.4 and 95.2%, respectively. Cells were loaded onto the 10X Genomics platform according to the

◄ **Figure 6. CrebA is transcriptionally repressed by FoxO in follicle cells.**

(A) Predicted FoxO binding sites, public ChIP-seq data and ChIP-qPCR primer designs at the *CrebA* gene locus. (B) Sequence logos of the consensus FoxO binding motif (JASPAR MA2236.1), and the two predicted 9mer sites (P1 and P1 in panel A). (C) ChIP-qPCR fold enrichment measured using the two primer pairs of panel A in samples from *tj>FoxO-GFP* and *tj>FoxO-GFP;InR^DN* ovaries. $N = 3$ biological replicates. Error bars represent mean ± SEM. Student's *t*-test $p = 0.02$ (denoted as *) and 0.04 (*), respectively. (D) RT-qPCR measures the ovarian level of *CrebA* mRNA in *tj > +*, *tj > FoxO-GFP* and *tj>FoxO-GFP;InR^DN* females. $N = 3$ biological replicates. Error bars represent mean ± SEM. One-way ANOVA $p = 0.55$ (denoted as ns) and < 0.0001 (****), respectively. (E) Representative images show the expression of FoxO-GFP (green) and CrebA (red) in main-body follicle cells at stages 8–11 from *tj>FoxO-GFP* females. Nuclei were counterstained with DAPI (blue). Scale bars = 20 μm. (F) Nuclear-to-cytoplasmic ratios of FoxO-GFP and CrebA proteins quantified from panel E. $N = 15$, 21, 20, and 15 follicle cells, respectively. Error bars represent mean ± SEM. One-way ANOVA was performed to compare each stage with stage 8. For CrebA, all *p* values <0.0001 (denoted as ****); for FoxO-GFP, $p < 0.0001$ (****), <0.0001 (****) and =0.58 (ns), respectively. (G) Representative images show the expression of FoxO-GFP (green) and CrebA (red) in follicle cells at stage 9 from *tj>FoxO-GFP;InR^DN* females. Scale bar = 20 μm. Source data are available online for this figure.

manufacturer's instructions. The cDNA libraries were generated with the Single Cell 3' Library & Gel Bead Kit v3 and sequenced with NovaSeq6000.

## Analysis of ovarian scRNA-seq data

The scRNA-seq outputs were processed as previously described (Yin, Ding et al, 2023). Then the two high-quality single-cell Seurat objects were integrated with Harmony v1 (Qiu, Mao et al, 2017). Eighteen clusters were decided as the final set of unique cell types based on the expression distributions of published gene markers (Jevitt et al, 2020), and their most representative gene markers are shown in Fig. EV2. For pseudotemporal trajectory analysis, the cells of interest were re-clustered with Monocle v2 (Korsunsky, Millard et al, 2019). Single-cell transcription regulatory networks were inferred and analyzed using SCENIC v0.9.1 with *Drosophila*-specific parameters (Aibar et al, 2017).

## Bioinformatics analyses of DNA and protein sequences

Two datasets of FoxO ChIP-seq are publicly available in the ENCODE portal, ENCFF899XPB and ENCFF014ZWS, which were generated using anti-GFP on mixed-sex wandering larvae expressing FoxO-GFP (Kudron et al, 2018; Data ref: Kudron et al, 2018). According to their analysis, two peaks were called out at the *CrebA* locus, −1931 to −1552bp and −801 to −422bp upstream of the *CrebA* gene, respectively.

The binding motif of FoxO and its letter-probability matrix (MA2236.1) were obtained from JASPAR (Rauluseviciute, Riudavets-Puig et al, 2023). FIMO from the MEME suite was used to scan the CrebA gene and its upstream 3 kb for this motif (Grant, Bailey et al, 2011). A total of 18 putative sites were found with a score >10 and *p* value <0.001, 16 in the gene body and two in the upstream sequence. Each of these two upstream sites lies within one of the two aforementioned peak regions, respectively. The sequence logos were generated with ggseqlogo (Wagih, 2017).

Peptide sequences of the bZIP domains of CrebA and CREB3L1-4 were obtained from UniProt. The alignment and phylogenetic tree construction were performed with web-based tools of UniProt.

## Chromatin immunoprecipitation qPCR

Freshly dissected ovaries were fixed with 1% formaldehyde for 10 min, washed once with PBS, and then transferred on ice with the addition of 250 mM glycine for 5 min to stop crosslinking. Subsequent steps followed the instructions of ChIP-IT Express Enzymatic (Active Motif). The DNA was extracted with the ChromoTek GFP-Trap Magnetic Agarose (Proteintech) beads, and purified with the FastPure Cell/Tissue DNA Isolation Mini Kit (Vazyme). The qPCR primers are listed in the Reagents and tools table. The qPCR results were normalized using the fold enrichment method as: $\log_2(\text{fold enrichment}) = \Delta Ct^{IgG} - \Delta Ct^{normalized\ ChIP}$.

## RT-qPCR

Total RNA were extracted with RNAiso Plus (Takara). Total RNA was reverse transcribed into first-strand cDNA using the ABScript II RT Master Mix for qPCR (ABclonal). qPCR was performed with SYBR Green Fast qPCR Mix (ABclonal) on QuantGene 9600 (Bioer). The primers are listed in the Reagents and tools table.

## Human cell culture

KGN cells (Procell) were cultured in DMEM/F12 medium (Servicebio) supplemented with 10% FBS and 100 U/mL of penicillin/streptomycin (Biosharp) at 37 °C in a humidified atmosphere with 5% $CO_2$. For knockdown experiments, siRNA transfection was conducted using jetPRIME Versatile DNA/siRNA transfection reagent (Polyplus). Specific siRNAs and their scrambled control oligos were synthesized by GenePharma. The sequences are listed in the Reagents and tools table. For Western blot analysis, rabbit anti-CYP19A1 (Proteintech, 1:500) and HRP-conjugated β-tubulin mouse mAb (ABclonal, 1:10000) were used.

## Estradiol ELISA

Cells were seeded in six-well plates at a density of $2 \times 10^5$ per well and cultured for 24 h before siRNA transfection. After 6 h of transfection, the medium was replaced with fresh culture medium. Following an additional 42 h of incubation, the culture supernatant was collected, and the 17β-estradiol concentration was measured using an E2 ELISA Kit (ABclonal).

## Statistics

Statistical analyses were performed using GraphPad Prism 9. Pairwise comparisons were performed using two-tailed Student's *t*-tests. For comparisons among more than two groups, we performed analysis of variance (ANOVA) for the overall test, followed by the Games–Howell post hoc test and Bonferroni correction. *N* denotes the number of samples or biological replicates; *ns*, *, **, ***, and **** denote *p* values >0.05 (testing was performed but the difference was not significant), 0.01–0.05, 0.001–0.01, 0.0001–0.001,

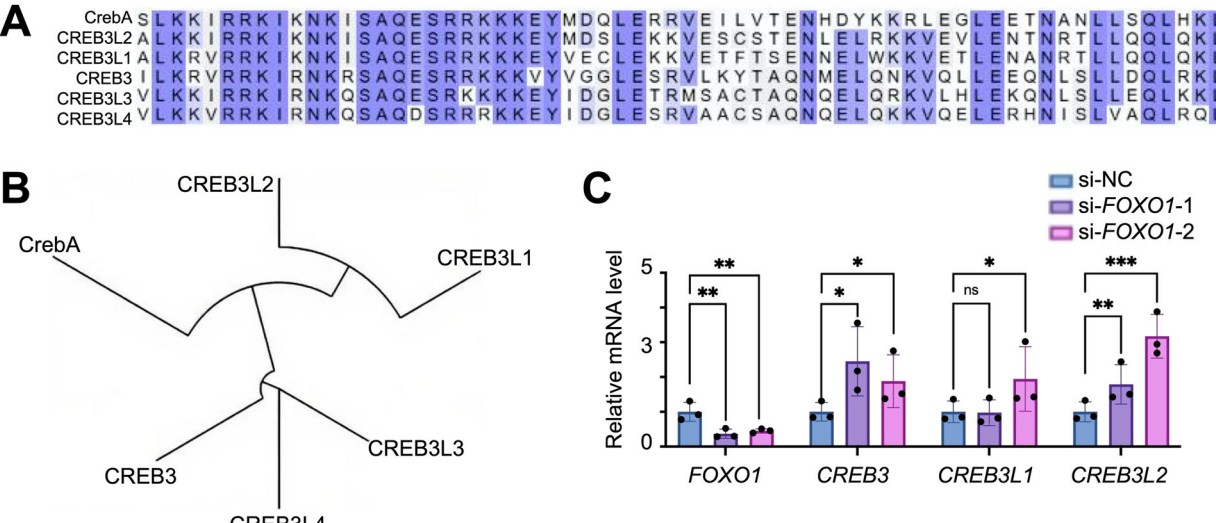

**D** Top 10 GO/KEGG enriched with genes downregulated by *CREB3L2* RNAi

| GO/KEGG terms | Count | Benjamini |
|---|---|---|
| extracellular region | 42 | 1.0E-12 |
| female pregnancy | 10 | 1.8E-07 |
| extracellular matrix structural constituent | 8 | 7.0E-04 |
| collagen-containing extracellular matrix | 11 | 1.1E-03 |
| cell surface | 13 | 2.6E-03 |
| extracellular matrix | 8 | 2.8E-03 |
| Cytokine-cytokine receptor interaction | 11 | 3.0E-03 |
| hormone activity | 7 | 3.2E-03 |
| PI3K-Akt signaling pathway | 11 | 5.3E-03 |
| JAK-STAT signaling pathway | 8 | 5.3E-03 |

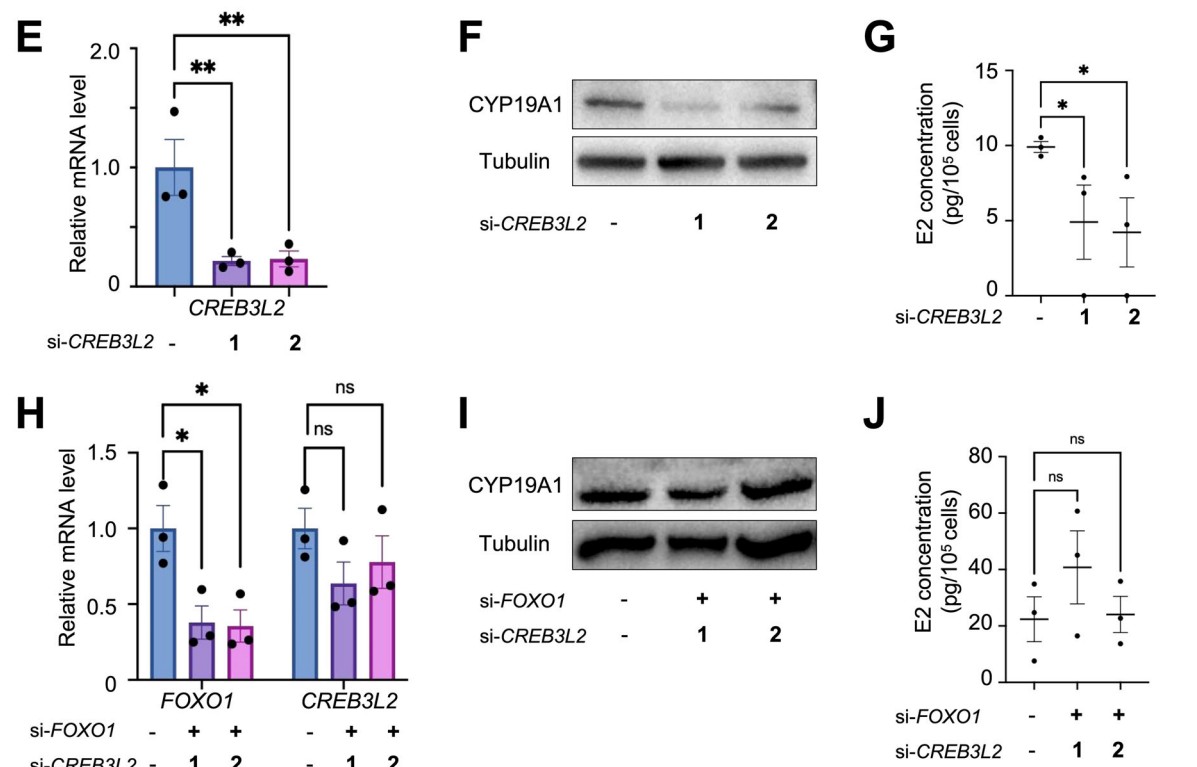

◄ **Figure 7. CREB3L2 regulates secretion and E2 activity in human KGN cells.**

(A) Alignment of the bZIP domains of CrebA, CREB3, and CREB3L1-4. Within this domain, CREB3 and CREB3L1-4 share 53.1, 60.9, 70.3, 54.7, and 51.6% identities with CrebA, respectively. (B) Phylogenetic tree based on the alignment of the bZIP domains. (C) RT-qPCR shows the increases of *CREB3*, *CREB3L1* and *CREB3L2* mRNA in KGN cells treated with *FOXO1* siRNAs. $N = 3$ biological replicates for each experimental group. Error bars represent mean ± SEM. One-way ANOVA *p* values = 0.005 (denoted as **), 0.01 (**), 0.02 (*), 0.047 (*), 0.93 (ns), 0.03 (*), 0.008 (**), and 0.0002 (***) for the significance marks from left to right respectively. (D) GO/KEGG terms enriched with genes that were downregulated in KGN cells treated with a *CREB3L2* siRNA. (E–G) RT-qPCR, Western blot and ELISA analyses show that both CYP19A1 protein and 17-beta-estradiol (E2) were reduced by *CREB3L2* siRNA treatments. $N = 3$ biological replicates for each experiment. Error bars represent mean ± SEM. One-way ANOVA *p* = 0.008 (**) and 0.009 (**) for qPCR, and *p* = 0.04 (*) and 0.04 (*) for ELISA, respectively. (H–J) RT-qPCR, Western blot and ELISA analyses show that knockdown of *FOXO1* counteracted the effects of *CREB3L2* siRNAs on *CREB3L2* mRNA, CYP19A1 protein and E2 levels. $N = 3$ biological replicates for each experiment. Error bars represent mean ± SEM. One-way ANOVA *p* = 0.01 (*), 0.01 (*), 0.12 (ns), and 0.34 (ns) for qPCR, and *p* = 0.06 (ns) and 0.84 (ns) for ELISA, respectively. Source data are available online for this figure.

and <0.0001, respectively; error bars represent mean ± standard error of the mean (SEM).

## Data availability

All raw RNA-seq data generated in this study have been deposited to the National Center for Biotechnology Information database under accession number PRJNA1288688.

The source data of this paper are collected in the following database record: biostudies:S-SCDT-10_1038-S44319-025-00672-6.

## Peer review information

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

## Acknowledgements

We thank Drs. Xiaohang Yang and Hai Huang at Zhejiang University and Dr. Yan Yan at Hong Kong University of Science and Technology for fly strains. We thank the Analysis Center of Agrobiology and Environmental Sciences at Zhejiang University for technical support of transmission electron microscopy. We acknowledge support from the School of Medicine, the affiliated Women's Hospital, and Zhejiang University. This study was supported by the National Natural Science Foundation of China (32470584 to F.H.) and the National Key R&D Program of China (2021YFC2700403 to J.M. and F.H.).

## Author contributions

**Xiaoya Wang**: Conceptualization; Data curation; Formal analysis; Validation; Investigation; Visualization. **Huanju Liu**: Conceptualization; Data curation; Formal analysis; Investigation; Visualization; Writing—original draft. **Zhiyong Yin**: Data curation; Software; Formal analysis; Visualization. **Tianning Shao**: Investigation. **Lin Li**: Investigation. **Jun Ma**: Conceptualization; Resources; Supervision; Funding acquisition; Project administration; Writing—review and editing. **Feng He**: Conceptualization; Data curation; Formal analysis; Funding acquisition; Validation; Investigation; Visualization; Writing—original draft; Project administration; Writing—review and editing.

Source data underlying figure panels in this paper may have individual authorship assigned. Where available, figure panel/source data authorship is listed in the following database record: biostudies:S-SCDT-10_1038-S44319-025-00672-6.

## Disclosure and competing interests statement

The authors declare no competing interests.

# Expanded View Figures

**Figure EV1. InR activity in stretched cells and centripetal cells.**

(A) Quantification of InR activity in stretched cells (blue rectangle) at stage 9. $N = 6$ egg chambers. The quantification was performed using the same method as in Fig. 1B. Error bars represent mean ± SEM. Scale bar = 100 μm. (B) Quantification of InR activity in stretched cells at stage 10. $N = 5$ egg chambers. Error bars represent mean ± SEM. Scale bar = 100 μm. (C) For centripetal cells, the quantification was performed within the border region of a $z$-stacked image (a total of 33 slices and the interval between neighboring slices = 2 μm). $N = 7$ egg chambers. Error bars represent mean ± SEM. Scale bar = 100 μm. We speculate that the interiorly situated cells might have an insufficient accessibility to insulin-like peptides, a possibility worthy for future investigation. Source data are available online for this figure.

▶

**A**

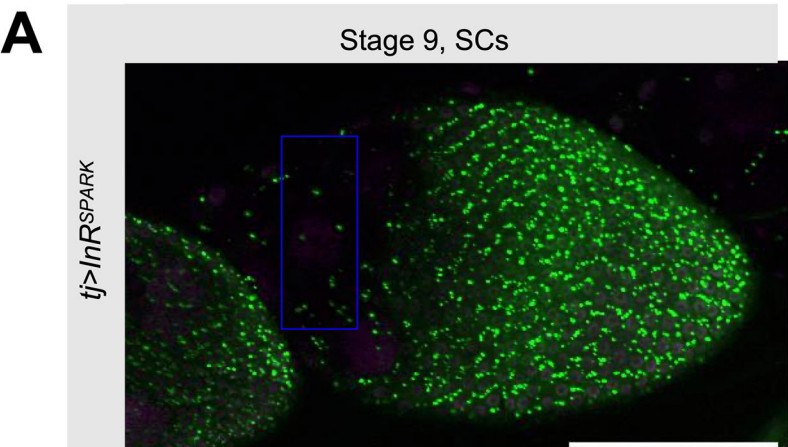

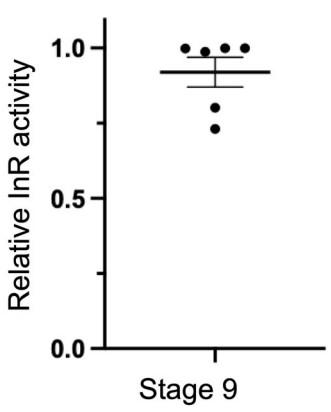

**B**

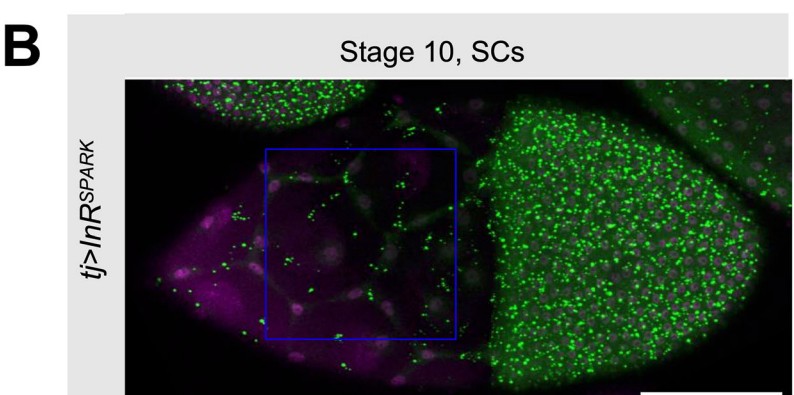

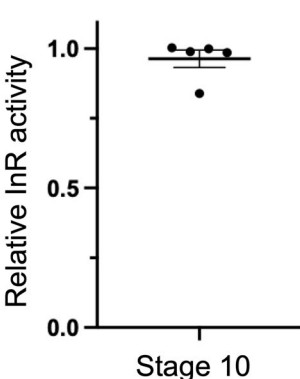

**C**

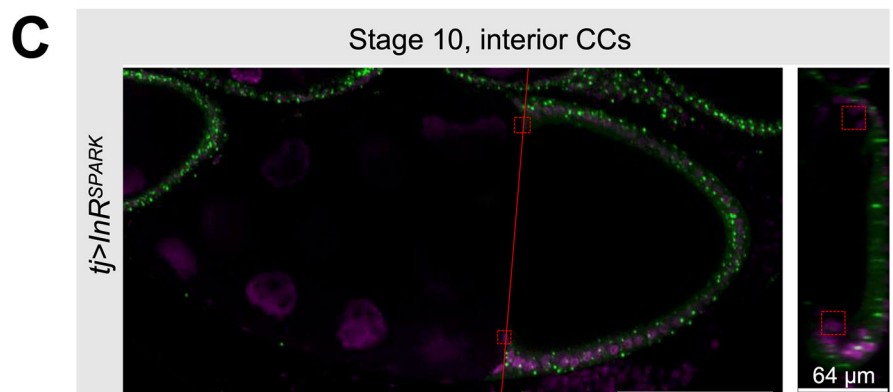

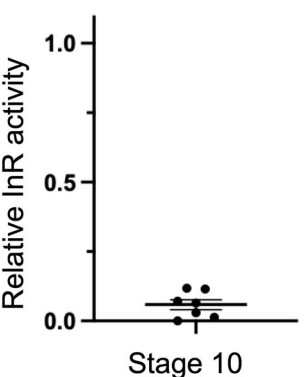

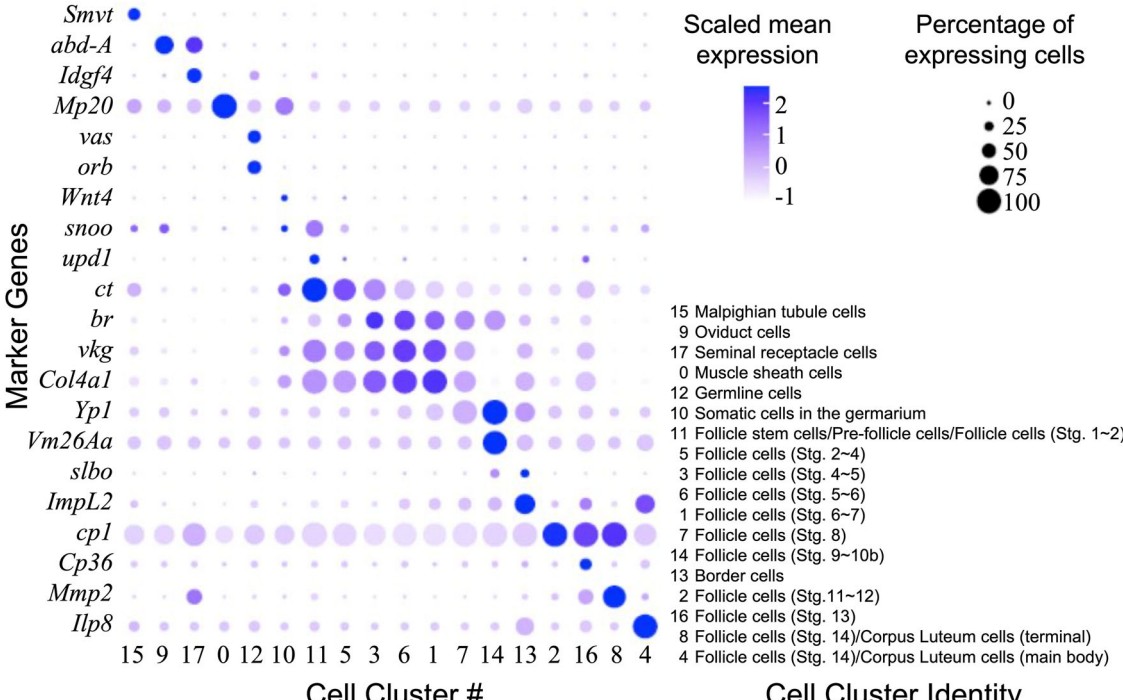

**Figure EV2. Marker genes to annotate cell clusters from scRNA-seq data.**

A dotplot of the scaled expression of marker genes in each inferred cell type. The size of each dot represents the percentage of cells in a given cluster expressing a given gene, and the color of each dot represents a *z*-score-scaled value of the gene's average expression across all cells in the cluster. Source data are available online for this figure.

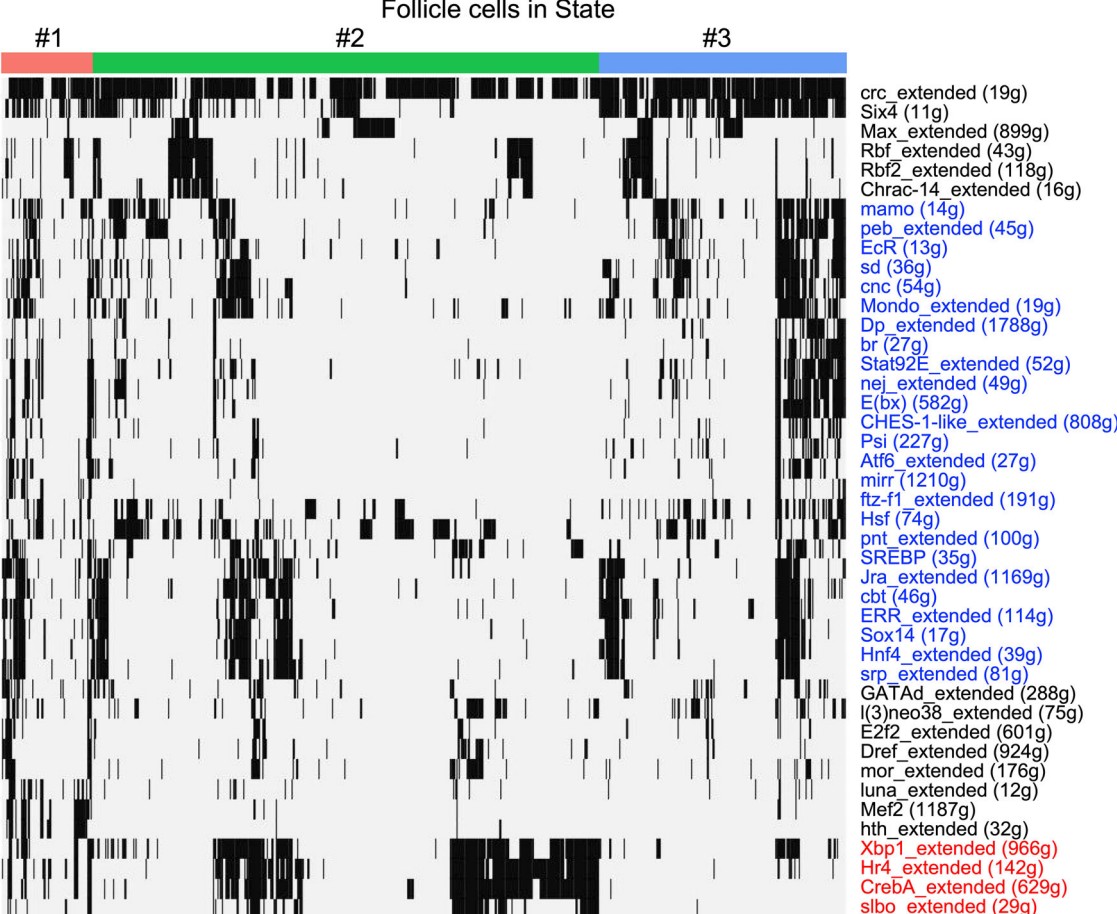

Transcriptional regulons highly expressed in follicle cells of Stg. 8~10B

Figure EV3. Active transcriptional regulons in vitellogenic follicle cells.

SCENIC (Van de Sande et al, 2020) was used to infer the single-cell transcription regulatory network. This figure plots the binary activity scores of all the 42 regulons identified from follicle cells at stages 8–10b. A black bar indicates an active state of the given regulon in the given cell. Grouped by hierarchical clustering, 24 regulons have elevated activities in the HSD-induced population #3 (blue), and 4 regulons show deactivation in this population (red). Source data are available online for this figure.

**A**

### Top 10 GO terms enriched with CrebA-regulon genes

| GO term | Count | Benjamini |
|---|---|---|
| COPII-coated ER to Golgi transport vesicle | 12 | 2.3E-08 |
| membrane | 184 | 6.9E-07 |
| COPII vesicle coat | 6 | 3.8E-05 |
| regulation of lipid storage | 9 | 8.7E-04 |
| signal peptidase complex | 4 | 6.8E-03 |
| positive regulation of cell migration | 7 | 1.0E-02 |
| SRP-dependent cotranslational protein targeting to membrane, translocation | 5 | 1.0E-02 |
| positive regulation of transcription by RNA polymerase II | 33 | 1.2E-02 |
| vitelline membrane formation involved in chorion-containing eggshell formation | 6 | 1.5E-02 |
| oligosaccharyltransferase complex | 5 | 1.8E-02 |

**B**

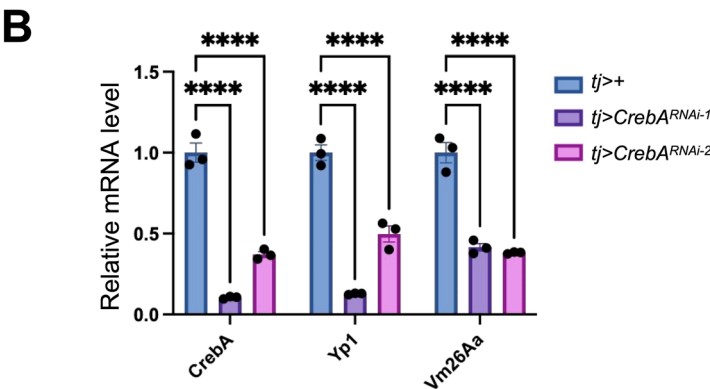

Figure EV4.  **CrebA is required for ovarian transcription of yolk protein and vitelline membrane genes.**

(A) Top ten GO terms enriched with CrebA-regulon genes. (B) RT-qPCR analysis of the ovarian mRNA levels of *CrebA*, *Yp1*, and *Vm26Aa* in *tj> CrebA^RNAi-1* and *tj> CrebA^RNAi-2* females. For each genotype, seven pairs of ovaries were sampled. Data were presented as mean ± SEM (*N* = 3 biological replicates for each measurement). Error bars represent mean ± SEM. One-way ANOVA was performed to compare either RNAi line with the control line, and all *p* values <0.0001 (denoted as ****). Source data are available online for this figure.

**A**

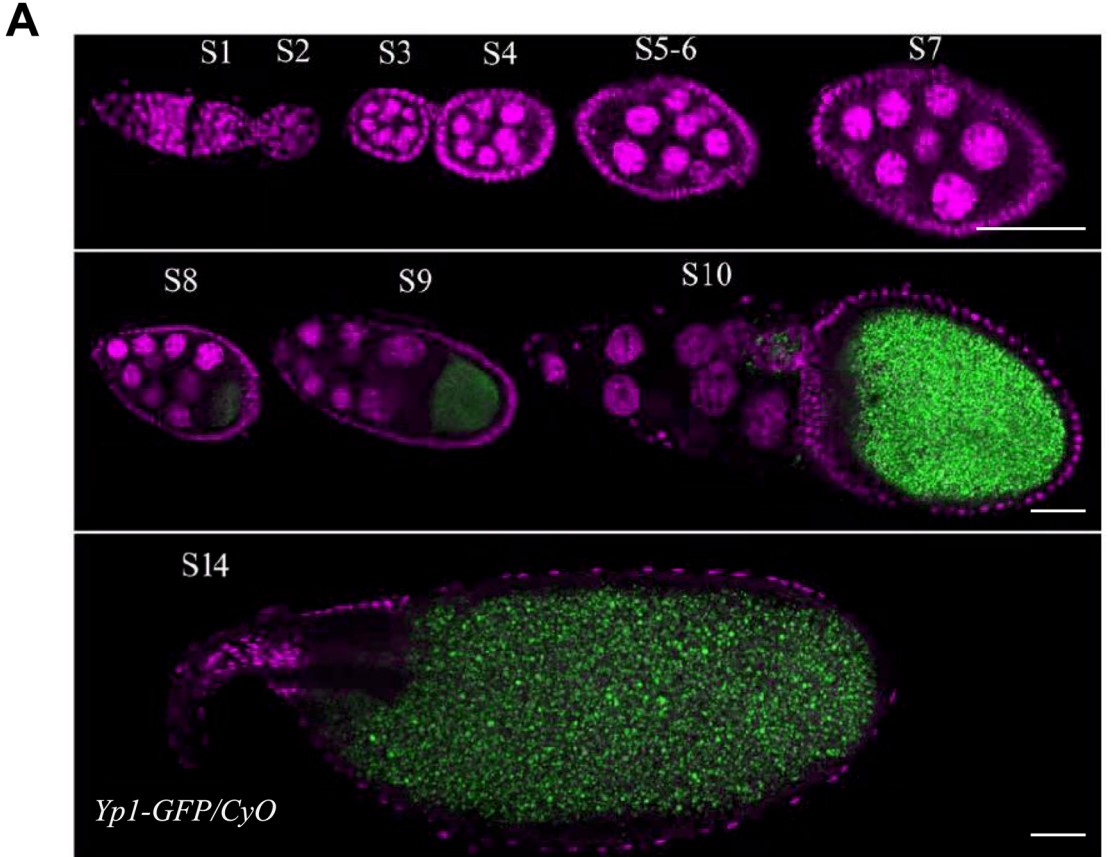

*Yp1-GFP/CyO*

**B**

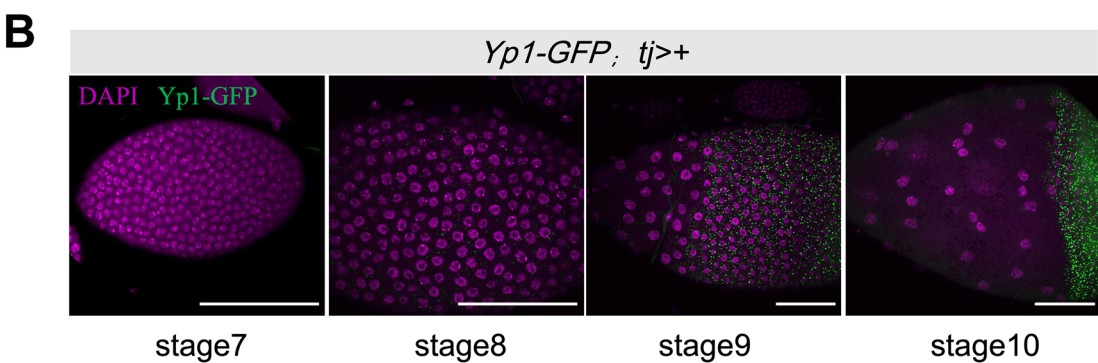

*Yp1-GFP；tj>+*

DAPI Yp1-GFP

stage7　　　stage8　　　stage9　　　stage10

**C**

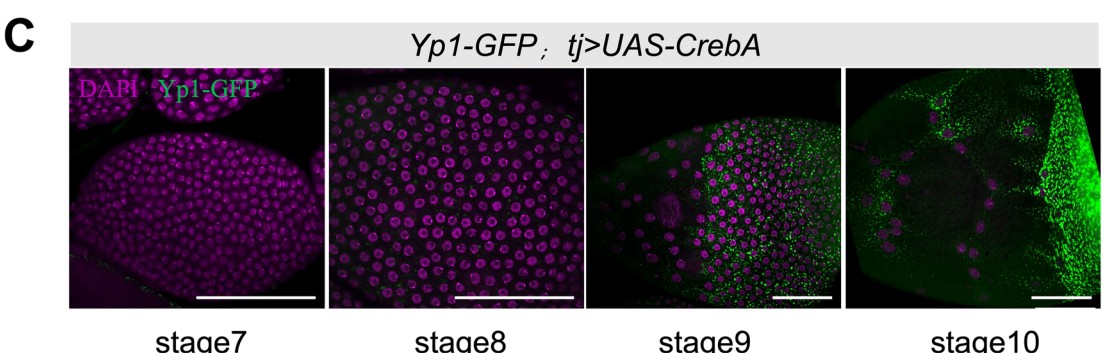

*Yp1-GFP；tj>UAS-CrebA*

DAPI Yp1-GFP

stage7　　　stage8　　　stage9　　　stage10

◀ **Figure EV5. CrebA can ectopically activate the expression of Yp1-GFP in follicle cells.**

(**A**) Representative images of Yp1-GFP (green) in egg chambers at different stages from *Yp1-GFP/CyO* females. Nuclei were counterstained with DAPI (magenta). Scale bars = 50 μm. (**B**) Representative images of Yp1-GFP in egg chambers at different stages from *Yp1-GFP/tj-Gal4;UAS-CrebA/+* females. Scale bars = 50 μm. (**C**) Representative images of Yp1-GFP in egg chambers at different stages from *Yp1-GFP/tj-Gal4;UAS-CrebA/+* females. Scale bars = 50 μm. We note that, in stretched cells where both CrebA and Yp1-GFP are normally low, ectopic expression of CrebA significantly increased Yp1-GFP. Source data are available online for this figure.

