## [Peer Review File · EMBO Reports]

An insulin receptor activity surge in follicle cells drives vitellogenesis by upregulating CrebA

Xiaoya Wang, Huanju Liu, Zhiyong Yin, Tianning Shao, Lin Li, Jun Ma, and Feng He

Corresponding author(s): Feng He (feng_he@zju.edu.cn) , Jun Ma (jun_ma@zju.edu.cn)

Review Timeline:

Submission Date:	6th Aug 25
Editorial Decision:	19th Sep 25
Revision Received:	20th Oct 25
Editorial Decision:	19th Nov 25
Revision Received:	24th Nov 25
Accepted:	3rd Dec 25

Editor: Achim Breiling

Transaction Report:

Dear Dr. He

Thank you for the submission of your research manuscript to our journal. Since my colleague Achim Breiling is currently out of office I have temporarily taken over the handling of your manuscript. We have meanwhile received the full set of referee reports that is copied below.

As you will see, the referees acknowledge that the findings are interesting and that the conclusions are overall supported by the data presented but they also raise a number of concerns and have suggestions how to further strengthen the data that need to be addressed.

Given these constructive comments, we would like to invite you to revise your manuscript with the understanding that the referee concerns (as detailed above and in their reports) must be fully addressed and their suggestions taken on board. Please address all referee concerns in a complete point-by-point response. Acceptance of the manuscript will depend on a positive outcome of a second round of review. It is EMBO Reports policy to allow a single round of revision only and acceptance or rejection of the manuscript will therefore depend on the completeness of your responses included in the next, final version of the manuscript.

We realize that it is difficult to revise to a specific deadline. In the interest of protecting the conceptual advance provided by the work, we recommend a revision within 3 months (December 18th). Please discuss the revision progress ahead of this time with the editor if you require more time to complete the revisions.

I am also happy to discuss the revision further via e-mail or a video call, if you wish.

=====

IMPORTANT NOTE:

We perform an initial quality control of all revised manuscripts before re-review. Your manuscript will FAIL this control and the handling will be delayed IN CASE the following APPLIES:

- 1) A data availability section providing access to data deposited in public databases is missing. If you have not deposited any data, please add a sentence to the data availability section that explains that.
- 2) Your manuscript contains statistics and error bars based on $n=2$. Please use scatter blots in these cases. No statistics should be calculated if $n=2$.

=====

- 2) individual production quality figure files as .eps, .tif, .jpg (one file per figure).

Please download our Figure Preparation Guidelines (figure preparation pdf) from our Author Guidelines pages <https://www.embopress.org/page/journal/14693178/authorguide> for more info on how to prepare your figures.

- 4) a complete author checklist, which you can download from our author guidelines (<<https://www.embopress.org/page/journal/14693178/authorguide>>). Please insert information in the checklist that is also reflected in the manuscript. The completed author checklist will also be part of the RPF.
- 5) Please note that all corresponding authors are required to supply an ORCID ID for their name upon submission of a revised manuscript (<<https://orcid.org/>>). Please find instructions on how to link your ORCID ID to your account in our manuscript tracking system in our Author guidelines (<<https://www.embopress.org/page/journal/14693178/authorguide#authorshipguidelines>>)
- 6) We replaced Supplementary Information with Expanded View (EV) Figures and Tables that are collapsible/expandable online. A maximum of 5 EV Figures can be typeset. EV Figures should be cited as 'Figure EV1, Figure EV2' etc... in the text and their respective legends should be included in the main text after the legends of regular figures.
- For the figures that you do NOT wish to display as Expanded View figures, they should be bundled together with their legends in a single PDF file called *Appendix*, which should start with a short Table of Content. Appendix figures should be referred to in the main text as: "Appendix Figure S1, Appendix Figure S2" etc. See detailed instructions regarding expanded view here: <<https://www.embopress.org/page/journal/14693178/authorguide#expandedview>>
 - Additional Tables/Datasets should be labeled and referred to as Table EV1, Dataset EV1, etc. Legends have to be provided in a separate tab in case of .xls files. Alternatively, the legend can be supplied as a separate text file (README) and zipped together with the Table/Dataset file.
- 7) Please note that a Data Availability section at the end of Materials and Methods is now mandatory. I note that you have already such a section, but we would need an URL that links directly to the dataset deposited at NCBI.
- 8) At EMBO Press we ask authors to provide source data for the main manuscript figures. You will receive a separate email with instructions for providing source data with your revised manuscript, including how to upload and organize the files.

Additional information on source data and instruction on how to label the files are available <<https://www.embopress.org/page/journal/14693178/authorguide#sourcedata>>

10) Figure legends and data quantification:
The following points must be specified in each figure legend:

- the name of the statistical test used to generate error bars and P values,
 - the EXACT p-values,
 - the number (n) of independent experiments (please specify technical or biological replicates) underlying each data point,
 - the nature of the bars and error bars (s.d., s.e.m.)
- If the data are obtained from n {less than or equal to} 5, show the individual data points in addition to the SD or SEM.
 - If the data are obtained from n {less than or equal to} 2, use scatter blots showing the individual data points.

See also the guidelines for figure legend preparation:
<https://www.embopress.org/page/journal/14693178/authorguide#figureformat>

11) Our journal encourages inclusion of *data citations in the reference list* to directly cite datasets that were re-used and obtained from public databases. Referee #3's suggestion regarding Figure 6A could easily be addressed using this format.

Data citations in the article text are distinct from normal bibliographical citations and should directly link to the database records from which the data can be accessed. In the main text, data citations are formatted as follows: "Data ref: Smith et al, 2001" or "Data ref: NCBI Sequence Read Archive PRJNA342805, 2017". In the Reference list, data citations must be labeled with "[DATASET]". A data reference must provide the database name, accession number/identifiers and a resolvable link to the landing page from which the data can be accessed at the end of the reference. Further instructions are available at <<https://www.embopress.org/page/journal/14693178/authorguide#referencesformat>>.

12) All Materials and Methods need to be described in the main text using our 'Structured Methods' format. According to this format, the Methods section includes a Reagents and Tools Table (listing key reagents, experimental models, software and relevant equipment and including their sources and relevant identifiers) followed by a Methods and Protocols section describing the methods, ideally using a step-by-step protocol format. The aim is to facilitate adoption of the methodologies across labs. Please download and fill our Reagents and Tools Table template (.docx), which you can find in our author guidelines: <https://www.embopress.org/page/journal/14693178/authorguide#structuredmethods>.

13) As part of the EMBO publication's Transparent Editorial Process, EMBO Reports publishes online a Review Process File to accompany accepted manuscripts. This File will be published in conjunction with your paper and will include the referee reports, your point-by-point response and all pertinent correspondence relating to the manuscript.

Yours sincerely,

=====

Referee #1:

In this study, Wang et al. identify a regulatory network involving insulin signaling driving CrebA expression which promotes yolk protein expression and vitelline membrane formation in *Drosophila* follicle cells during oogenesis.

They claim to have identified that:

1. Insulin receptor (InR) kinase activity increases in follicle cells during vitellogenic stages (stage 8-10 egg chambers).
2. The increase in InR kinase activity drives expression of the transcription factor CrebA, which promotes yolk protein biosynthesis and vitelline membrane formation, which are both necessary for maturation of the oocyte.
3. InR kinase activity negatively regulates FoxO-mediated repression of CrebA.
4. This gene regulatory axis (FoxO repression of CrebA) is conserved in humans, where CREB3L2 depletion leads to a decrease in secretory protein expression and insulin signaling.

Conceptual advance: It is known in humans that maternal weight/diet can affect pregnancy and viability of the fetus. It is also known in *Drosophila* oogenesis that insulin signaling promotes germline development at early stages and folliculogenesis at later (stage 8+) stages. It has not yet been shown how insulin signaling affects late stage folliculogenesis. This paper offers a conceptual advance in bridging this knowledge gap by reporting that a transcription factor, CrebA, is a key mediator of InR activity in vitellogenic follicle cells. The authors also show that CrebA activity is inhibited by FoxO, likely by binding upstream of CrebA to repress its expression, which is novel.

Overall, this paper is well done, the data substantiated, and well-presented with a only few suggestions that are listed below:

Major problems with data:

1. Inadequate comparison of different follicle cell types
 - a. The authors first analyzed InR activity using the phase separation-based EGFP tool across different follicle cell types. However, the rest of the study does not delineate between these different cell types, which could be relevant since the InRSPARK activity does not increase in centripetal cells. The rest of the paper does not compare CrebA expression or FoxO-

GFP expression across these different follicle cell types. They should mention if this is consistent across all cell types.

2. CrebA expression in the follicle cells is not clear in Fig 3D

a. The authors show immunostaining data in Fig 3D to validate lower protein expression of CrebA in stage-9 follicle cell nuclei in HSD compared to ND females. The fluorescence intensity of DAPI is low in the HSD image compared to the ND image, complicating any fluorescence intensity comparisons of CrebA across the 2 conditions. Quantification with a normalizing gene would be beneficial for Fig 3D.

b. Additionally, could a western blot have been performed to compare protein levels?

3. Dose-dependent effect of ovarian CrebA

a. The authors report that certain CrebA target genes, including CrebA, are more prominently reduced in *tj>CrebARNai-1* compared to *tj>CrebARNai-2*, and state that this suggests a dose-dependent effect of ovarian CrebA. It is unclear what the difference is across the 2 RNAi lines, and why this would indicate a dose-dependent effect.

4. While the paper is well written, there could be a few changes:

a. The introduction could use a paragraph about HSD - when I was reading the results this came out of the blue for me.

b. References in the results section are sparse, for example, for all the RNA packages and Creb interactors etc.

Other comments:

1. Figure 6E: the text claims that FoxO-GFP is decreased in stage 9 & 10, however in the images it seems to be brighter in the follicle cell nuclei. Could the magnification be increased to better see the follicle cell nuclei?

2. Figure 1B, 1D and 1G: SPARK activity does not have units for the y-axis label

3. Page 4, line 23: "...peaked at vitellogenic stages 8~10 for MBFCs and SCs, not CCs, followed by a sharp..."

4. Page 5, line 4: "...reduced from stage 8 to 11~12 in females...."

5. Page 5, line 5: "...document a stage-specific/vitellogenic surge..." instead of "temporal"

6. Page 5, line 5: "...surge of InR activity in stage 8 follicle cells..."

7. Page 5, line 7: "...follicle cells is caused by HSD."

8. Page 8, line 13: Figure 4D is cited but it's Figure 4E.

9. Page 10, line 3: Figure 5I is cited but it's Figure 5C.

Referee #2:

This manuscript by Wang et al describes the functional analysis of a previously identified surge in insulin receptor signalling during folliculogenesis in the *Drosophila* ovary and provides robust evidence that an increase in transcription factor CrebA expression plays a key downstream function. The authors demonstrate a central role for CrebA using established markers for this protein to show upregulation in CrebA protein (and mRNA) levels between stages 9 and 10, and knockdown and overexpression of this gene to reveal its downstream effects on vitellogenesis and yolk protein production.

Overall, I thought this was a well written manuscript with a set of clear findings that may be relevant to human oogenesis, as evidenced by some experiments on cultured human granulosa cells at the end of the results section. The study highlights how specific follicle cells at a particular stage of oogenesis can be sensitive to insulin-like proteins that are present at all stages of oogenesis and identifies one key downstream effector, providing an example of how diet may directly affect egg production and fertility in females. This makes the work of significant interest to researchers studying reproduction, but also to those interested in the effects of diet and the regulation of metabolism *in vivo*.

I have a small number of suggestions for changes that I think should be considered prior to publication:

1. In Figure 1A, stage 10b, SCs panel, I got the impression that the SPARK reporter shows some droplet formation in the CCs, which does not fit with the description provided in the text. Could this be clarified?

2. Figure S8: It looks to me that the egg as well as the egg shell are abnormal following follicular CrebA knockdown; it should be mentioned that this phenotype is more than an eggshell defect

3. Figure 5A, 5B, 5D and 5I (as they appear on the current Figure 5): I think there should be a CrebA overexpression control in ND (A) or in the absence of InR-DN (B, D, I) to determine to what extent CrebA can mediate its effects in a dominant way and hyper-activate the vitellogenic/yolk protein pathways, and what the impact is on egg laying. This is included in Figure 5I for diet where it suggests that the vitelline membrane can become thicker than normal and it would be interesting to see whether this affects egg-laying (Figure 5A). Also in figure 5I, the key comparison is HSD versus HSD + CrebA, but the significance of the rescue has not been marked on the bar chart.

4. The order of panels in Figure 5 has clearly been changed in this version of the manuscript, but the description of Figure 5 in the text has not (or vice versa), so Figure 5C is described in the text as Figure 5I, and Figure 5D is Figure 5C in the text, etc. The references to specific panels in the text and Figure need to be aligned.

5. In the Materials and Methods, the fly stocks are given as BL (Bloomington?) and THU numbers - what is the latter and please spell out the sources somewhere?

6. For statistical analysis, it is suggested a t-test was used unless otherwise specified. Several of the analyses involve multiple comparisons and this would require ANOVA, if normally distributed, or a test like Kruskal-Wallis, if not. The correct analysis needs to be undertaken and may affect some of the p values, though in most cases it appears that the changes are sufficiently large and consistent that they will still be significant. There are also no n numbers included, which should be done in the figures/legends; it would, in fact, be very helpful to show the individual data points on the graphs to see the spread of data.

Referee #3:

In this manuscript, the authors focus on the role of insulin signaling in *Drosophila* follicle cells and link insulin signaling to the effects of a maternal high sugar diet. Using a special high tech GFP reporter, they demonstrate a surge in insulin signaling in follicle cells during vitellogenesis that can be disrupted by maternal high sucrose diet. They show that with a high sugar diet, a population of stage 8 follicle cells exhibit reduced CrebA dependent transcription of secretory machinery genes and of yolk and vitelline membrane protein genes. Their studies indicate that insulin signaling works through CrebA to boost genes encoding components of the secretory machinery and expression of genes encoding yolk and vitelline membrane proteins. They provide evidence that CrebA is repressed by activated nuclear FoxO. Importantly, they show that all the effects of the high sugar diet can be mimicked by RNAi knockdown of CrebA and can also be rescued by CrebA expression specifically in follicle cells. Finally, they provide some preliminary evidence that the same pathway may have a parallel function in the ovarian granulosa cells of humans (using a cell line in culture).

This is an extensive and beautiful body of work, but, in my opinion, it needs a few tweaks before publication.

Issues to address:

In the abstract, would be good to use the word "boost" rather than "control" the secretory capacity of follicle cells. I think it's okay to reveal more about the findings in the abstract.

Figure 1: This figure introduces SPARK - phase separated EGFP reporter on kinase activity of InR and show that it's signal is dependent on InR signaling and a normal diet. The quantification in B regarding where SPARK is observed does not really match the images shown in A. For example, it's very hard to detect any signal in the stretch cells (blue boxes, blue line in graph), although based on the graph, that should be the highest signal.

Also, I can't see the absence of signal in centripetal cells - perhaps a larger magnification would make the signal versus background more obvious. I see from the materials and methods how they are quantifying the data, but that's not clear from the figure legend or the text describing the results from the figure. It needs to be described. They need to fix this - and either way, they should probably show some relative quantification of total and activated signal among the different follicular cell populations. Is the *tj-Gal4* line differentially expressed in the different follicular cell populations?

On the description of cluster results from scRNASeq of normal diet and high sugar diet ovaries, I didn't quite understand this sentence when I first read it: Unsupervised clustering analysis revealed a mixed distribution of ND and HSD cells in all clusters. I finally figured it out but the authors might find a simpler way to describe their findings that doesn't confuse the reader. I think what they mean is that they identified 17 cell clusters from their data and all clusters were populated both by normal diet and high sugar diet cells.

It was not clear how they assigned clusters to cell types until I looked at the supplemental data where I found the supplemental chart showing marker gene expression in the different clusters. The authors should mention how they did this in the main body of the text. I'm also guessing that the "marker genes" for each population were discovered by others and these people and their work should be acknowledged, even if the data are publicly available.

Figure 3B - I'm not sure where they came up with the pathway of cross-regulation between the four HSD-deactivated TFs they show in B. They should spell out where this pathway comes from or they should remove it.

What is CrebA-lacZ? The authors need to explain what this is - I had to look it up but it is essentially a reporter insertion line for assaying CrebA expression (and the authors should give some caveats regarding using this line since it may not fully report on all aspects of CrebA expression, given it's a P-element insertion line that disrupts CrebA function).

Figure 3D is not so convincing because not only is the green CrebA staining reduced with HSD, so is the purple DAPI nuclear staining and the DAPI looks to be reduced to a similar level as CrebA.

Figure 3 - The investigators need to provide more detail on how they found the "regulons" for each transcription factor they discuss.

Again in figure S3 - please explain how the "regulons" were discovered and what is being plotted.

Figure 4F - why does the VM with the CrebA knockout look more electron dense than in the control - Can the authors show a higher mag view? Might help shed light on the differences and help the reader see them a little more easily.

Figure 6A. Investigators need to acknowledge where the public ChIP-seq data for FoxO on the CrebA gene came from. Again, somebody did that work, even if it is publicly available. Also, please show the actual sequence of the two FoxO motifs. As a reader, I would like to see how well the sites in the CrebA upstream region match the defined consensus binding motif for FoxO and I don't want to have to look it up by downloading the CrebA enhancer region.

Bottom of page 12. The references for CrebA as a master regulator of the secretory machinery in embryonic and larval tissues are incorrect. Please remove the ones that are there and instead use: Abrams and Andrew, 2005; Fox et al., 2010; Johnson et al., 2020)

Figure 7. The homology between CrebA and the Creb3 family has been pointed out previously and extends into a region N-terminal to the bZIP DNA binding domain - the N-terminal region distinguishes this class of molecules from other Crebs. The authors should probably recognize the work that initially showed the extended homology and its importance in Creb3L function (Barbosa et al., 2013).

Figure S8 A - the *tj-Gal4*>+ embryo shown is a late stage compared to the "freshly laid" *tj-Gal4*>CrebARN*ai* embryo described. The authors should find and show a better stage comparison to match the CrebA knockdown embryo.

Page 8, the description of what they used to monitor Yp1 expression is not quite right - it needs to be reworded - change "expresses" to "is" should read: which is a large genomic clone that includes the Yp1 gene fused to green fluorescent protein tag".

Referee #1:

In this study, Wang et al. identify a regulatory network involving insulin signaling driving CrebA expression which promotes yolk protein expression and vitelline membrane formation in *Drosophila* follicle cells during oogenesis.

They claim to have identified that:

1. Insulin receptor (InR) kinase activity increases in follicle cells during vitellogenic stages (stage 8-10 egg chambers).
2. The increase in InR kinase activity drives expression of the transcription factor CrebA, which promotes yolk protein biosynthesis and vitelline membrane formation, which are both necessary for maturation of the oocyte.
3. InR kinase activity negatively regulates FoxO-mediated repression of CrebA.
4. This gene regulatory axis (FoxO repression of CrebA) is conserved in humans, where CREB3L2 depletion leads to a decrease in secretory protein expression and insulin signaling.

Conceptual advance: It is known in humans that maternal weight/diet can affect pregnancy and viability of the fetus. It is also known in *Drosophila* oogenesis that insulin signaling promotes germline development at early stages and folliculogenesis at later (stage 8+) stages. It has not yet been shown how insulin signaling affects late stage folliculogenesis. This paper offers a conceptual advance in bridging this knowledge gap by reporting that a transcription factor, CrebA, is a key mediator of InR activity in vitellogenic follicle cells. The authors also show that CrebA activity is inhibited by FoxO, likely by binding upstream of CrebA to repress its expression, which is novel.

Overall, this paper is well done, the data substantiated, and well-presented with only a few suggestions that are listed below:

Author response: We thank this reviewer for appreciating the quality of our work. We also thank this reviewer for making specific suggestions about improvement in the writing, and further analyses or experiments, which are now incorporated in our revised manuscript.

Major problems with data:

1. Inadequate comparison of different follicle cell types

a. The authors first analyzed InR activity using the phase separation-based EGFP tool across different follicle cell types. However, the rest of the study does not delineate between these different cell types, which could be relevant since the InRSPARK activity does not increase in centripetal cells. The rest of the paper does not compare CrebA expression or FoxO-GFP expression across these different follicle cell types. They should mention if this is consistent across all cell types.

Author response: We agree with this reviewer in that the current manuscript has a primary focus on main-body follicle cells. To avoid potential distractions, we have now moved the results of stretched cells and centripetal cells to Figure EV1. In addition, we have also provided a statement about the cell-type specificity of CrebA as shown in Figure EV5C, noting that the CrebA protein was much lower in stretched cells than that in main-body follicle cells.

2. CrebA expression in the follicle cells is not clear in Fig 3D

a. The authors show immunostaining data in Fig 3D to validate lower protein expression of

CrebA in stage-9 follicle cell nuclei in HSD compared to ND females. The fluorescence intensity of DAPI is low in the HSD image compared to the ND image, complicating any fluorescence intensity comparisons of CrebA across the 2 conditions. Quantification with a normalizing gene would be beneficial for Fig 3D.

b. Additionally, could a western blot have been performed to compare protein levels?

Author response: Thank the reviewer for suggesting important experiments to help address the raised concern. Because the females on high-sucrose diet (HSD) produced very few egg chambers beyond stage 8, we were constrained by a limited number of good-quality images of egg chambers at stage 9 for the initial submission. It is important to note that all the samples in the experiments for regular diet and HSD were prepared side by side, and their images were taken in a single operation cycle with consistent microscopic settings. To better illustrate the main conclusion for these results, we now present low-magnification images with more egg chambers, more clearly showing both the comparable DAPI intensities and the differences in CrebA between two diet conditions. In addition, we performed Western blot analysis shown in Figure 3D. Together, these results demonstrate a decrease of CrebA protein in ovaries of HSD females.

3. Dose-dependent effect of ovarian CrebA

a. The authors report that certain CrebA target genes, including CrebA, are more prominently reduced in *tj>CrebARNAi-1* compared to *tj>CrebARNAi-2*, and state that this suggests a dose-dependent effect of ovarian CrebA. It is unclear what the difference is across the 2 RNAi lines, and why this would indicate a dose-dependent effect.

Author response: Thank the reviewer for pointing this out. We agree that the “dose-dependent effect” in our initial writing was inaccurate and potentially misleading. In the revised manuscript, we simply state the fact that these molecular observations were consistent with the differences in phenotypic severity between the two RNAi lines.

4. While the paper is well written, there could be a few changes:

a. The introduction could use a paragraph about HSD - when I was reading the results this came out of the blue for me.

b. References in the results section are sparse, for example, for all the RNA packages and Creb interactors etc.

Author response: Thank the reviewer for these suggestions that have improved our writing. We have now included a description of the relationship between high-sucrose diet (HSD) and the insulin signaling pathway in the Introduction section. We have also incorporated more references for our methodology.

Other comments:

1. Figure 6E: the text claims that FoxO-GFP is decreased in stage 9 & 10, however in the images it seems to be brighter in the follicle cell nuclei. Could the magnification be increased to better see the follicle cell nuclei?

Author response: We have now shown a magnified view of the images and the profiles of nuclear-to-cytoplasmic ratios of CrebA and FoxO-GFP in the revised Figure 6E-F.

2. Figure 1B, 1D and 1G: SPARK activity does not have units for the y-axis label

Author response: The previous y-axis label was inaccurate. Thank you for pointing this out. We have changed it to "Relative InR activity" and improved the description about the quantification method in the figure legend.

3. Page 4, line 23: "...peaked at vitellogenic stages 8~10 for MBFCs and SCs, not CCs, followed by a sharp..."

4. Page 5, line 4: "...reduced from stage 8 to 11~12 in females..."

5. Page 5, line 5: "...document a stage-specific/vitellogenic surge..." instead of "temporal"

6. Page 5, line 5: "...surge of InR activity in stage 8 follicle cells..."

7. Page 5, line 7: "...follicle cells is caused by HSD."

8. Page 8, line 13: Figure 4D is cited but it's Figure 4E.

9. Page 10, line 3: Figure 5I is cited but it's Figure 5C.

Author response: We have made changes accordingly. Thank the reviewer very much.

Referee #2:

This manuscript by Wang et al describes the functional analysis of a previously identified surge in insulin receptor signalling during folliculogenesis in the *Drosophila* ovary and provides robust evidence that an increase in transcription factor CrebA expression plays a key downstream function. The authors demonstrate a central role for CrebA using established markers for this protein to show upregulation in CrebA protein (and mRNA) levels between stages 9 and 10, and knockdown and overexpression of this gene to reveal its downstream effects on vitellogenesis and yolk protein production.

Overall, I thought this was a well written manuscript with a set of clear findings that may be relevant to human oogenesis, as evidenced by some experiments on cultured human granulosa cells at the end of the results section. The study highlights how specific follicle cells at a particular stage of oogenesis can be sensitive to insulin-like proteins that are present at all stages of oogenesis and identifies one key downstream effector, providing an example of how diet may directly affect egg production and fertility in females. This makes the work of significant interest to researchers studying reproduction, but also to those interested in the effects of diet and the regulation of metabolism *in vivo*.

Author response: We thank this reviewer for the appreciation of the quality and significance of our work.

I have a small number of suggestions for changes that I think should be considered prior to publication:

1. In Figure 1A, stage 10b, SCs panel, I got the impression that the SPARK reporter shows some droplet formation in the CCs, which does not fit with the description provided in the text. Could this be clarified?

Author response: We agree with this reviewer in that our initial presentation about the observations in centripetal cells was confusing and potentially misleading. To improve the clarity of the manuscript, we have now moved the analyses of centripetal cells and stretched cells to supplementary materials, so that the main text has a primary focus on main-body follicle cells. In Figure EV1, we show that there were indeed abundant InR-SPARK droplets

in cortical follicle cells at the border position, but the droplets were absent in the centripetal cells inside the egg chamber. We provide a possibility that, among others, these interiorly situated cells might have an insufficient accessibility to insulin-like peptides.

2. Figure S8: It looks to me that the egg as well as the egg shell are abnormal following follicular CrebA knockdown; it should be mentioned that this phenotype is more than an eggshell defect

Author response: We agree with this reviewer and have changed the related statement.

3. Figure 5A, 5B, 5D and 5I (as they appear on the current Figure 5): I think there should be a CrebA overexpression control in ND (A) or in the absence of InR-DN (B, D, I) to determine to what extent CrebA can mediate its effects in a dominant way and hyper-activate the vitellogenic/yolk protein pathways, and what the impact is on egg laying. This is included in Figure 5I for diet where it suggests that the vitelline membrane can become thicker than normal and it would be interesting to see whether this affects egg-laying (Figure 5A). Also in figure 5I, the key comparison is HSD versus HSD + CrebA, but the significance of the rescue has not been marked on the bar chart.

Author response: We thank this reviewer for making these important suggestions. The manuscript has now included the results of CrebA overexpression in Figure 5. In addition, we have incorporated the significance mark for the comparison between HSD and HSD + CrebA to Figure 5I, which indicates a clear rescue effect of CrebA overexpression in the HSD background.

4. The order of panels in Figure 5 has clearly been changed in this version of the manuscript, but the description of Figure 5 in the text has not (or vice versa), so Figure 5C is described in the text as Figure 5I, and Figure 5D is Figure 5C in the text, etc. The references to specific panels in the text and Figure need to be aligned.

Author response: Thank this reviewer for pointing this error. It has now been corrected.

5. In the Materials and Methods, the fly stocks are given as BL (Bloomington?) and THU numbers - what is the latter and please spell out the sources somewhere?

Author response: BL is from the Bloomington stock center and THU is Tsinghua University Fly Center. We have included a Reagent Table with all the related information.

6. For statistical analysis, it is suggested a t-test was used unless otherwise specified. Several of the analyses involve multiple comparisons and this would require ANOVA, if normally distributed, or a test like Kruskal-Wallis, if not. The correct analysis needs to be undertaken and may affect some of the p values, though in most cases it appears that the changes are sufficiently large and consistent that they will still be significant. There are also no n numbers included, which should be done in the figures/legends; it would, in fact, be very helpful to show the individual data points on the graphs to see the spread of data.

Author response: We thank this reviewer for making these specific suggestions on statistical analysis. We have made the following improvements in the revised manuscript. 1) To evaluate the pairwise differences across more than two experimental groups, we performed

one-way ANOVA for the overall test, followed by the Games-Howell post-hoc test and Bonferroni correction. By doing this, we found that several p -values were changed, but no conclusion was affected. We have included all the exact values for these significance levels in figure legends. 2) All numbers of samples or experiments are now reported in figure legends. 3) Individual data points are added onto some of the graphs.

Referee #3:

In this manuscript, the authors focus on the role of insulin signaling in *Drosophila* follicle cells and link insulin signaling to the effects of a maternal high sugar diet. Using a special high tech GFP reporter, they demonstrate a surge in insulin signaling in follicle cells during vitellogenesis that can be disrupted by maternal high sucrose diet. They show that with a high sugar diet, a population of stage 8 follicle cells exhibit reduced CrebA dependent transcription of secretory machinery genes and of yolk and vitelline membrane protein genes. Their studies indicate that insulin signaling works through CrebA to boost genes encoding components of the secretory machinery and expression of genes encoding yolk and vitelline membrane proteins. They provide evidence that CrebA is repressed by activated nuclear FoxO. Importantly, they show that all the effects of the high sugar diet can be mimicked by RNAi knockdown of CrebA and can also be rescued by CrebA expression specifically in follicle cells. Finally, they provide some preliminary evidence that the same pathway may have a parallel function in the ovarian granulosa cells of humans (using a cell line in culture).

This is an extensive and beautiful body of work, but, in my opinion, it needs a few tweaks before publication.

Author response: We thank this reviewer for appreciating our work and for making specific suggestions to help improve our manuscript.

Issues to address:

In the abstract, would be good to use the word "boost" rather than "control" the secretory capacity of follicle cells. I think it's okay to reveal more about the findings in the abstract.

Author response: Thank you for this suggestion. We have now used the word "boost" to better highlight the role of CrebA in elevating the secretory function of the ovary.

Figure 1: This figure introduces SPARK - phase separated EGFP reporter on kinase activity of InR and show that it's signal is dependent on InR signaling and a normal diet. The quantification in B regarding where SPARK is observed does not really match the images shown in A. For example, it's very hard to detect any signal in the stretch cells (blue boxes, blue line in graph), although based on the graph, that should be the highest signal. Also, I can't see the absence of signal in centripetal cells - perhaps a larger magnification would make the signal versus background more obvious. I see from the materials and methods how they are quantifying the data, but that's not clear from the figure legend or the text describing the results from the figure. It needs to be described. They need to fix this - and either way, they should probably show some relative quantification of total and activated signal among the different follicular cell populations.

Author response: We apologize to all the three reviewers for the confusion brought by our data presentation in the initial Figure 1. This confusion resulted from an insufficient

description of the quantification method and inadequate images to perform the cell-type specific analysis. To prevent this confusion, we have made the following improvements during the revision. 1) Because the current manuscript primarily focuses on main-body follicle cells but not the specialized types, we have moved the analysis on centripetal cells and stretched cells to supplementary materials to avoid potential distractions. 2) We have described the quantification method in the legend to Figure 1B. Alongside the panel of quantitative profiles, a magnified view is presented to show the droplets and the cell region from where fluorescent intensities are quantified as the signal and the background, respectively. 3) We show that the SPARK droplets are formed sharply in stretched cells in Figure EV1A-B. We suspect a dilution effect on the cytoplasmic concentration due to the enlarged cell size. 4) We show a transverse plane image at the border region, from which the centripetal cells inside the egg chamber have nearly none droplet. It is possible that these interiorly situated cells might have an insufficient accessibility to insulin-like peptides, a possibility worthy of future investigation.

Is the *tj-Gal4* line differentially expressed in the different follicular cell populations?

Author response: We thank this reviewer for raising this concern. According to our scRNA-seq data, the endogenous *traffic jam (tj)* gene is uniformly expressed in all follicle cell clusters before stage 14. Our *tj-Gal4*-driven UAS-GFP (data not shown in this manuscript) exhibits no obvious expression preference for time or location in the egg chambers after stage 7. Furthermore, we used *tj-Gal4* to drive InR-SPARK to measure the intracellular InR activity and to drive FoxO-GFP to measure the nuclear-to-cytoplasmic ratio of FoxO. In both cases, the measurement relies on post-translational events more than the protein expression level. Therefore, and importantly, the stage-specific surge of InR activity in main-body follicle cells revealed by our reporter is unlikely caused by the *tj-Gal4* driver. However, it does not explain the difference in the cytoplasmic background of InR-SPARK between different cell types. While our preliminary analysis suggests a potential contribution of the cell size, it provides a future direction for investigation that is beyond the scope of the current work.

On the description of cluster results from scRNASeq of normal diet and high sugar diet ovaries, I didn't quite understand this sentence when I first read it: Unsupervised clustering analysis revealed a mixed distribution of ND and HSD cells in all clusters. I finally figured it out but the authors might find a simpler way to describe their findings that doesn't confuse the reader. I think what they mean is that they identified 17 cell clusters from their data and all clusters were populated both by normal diet and high sugar diet cells.

Author response: We thank this reviewer for offering such kindness. The mentioned sentence has been revised as suggested.

It was not clear how they assigned clusters to cell types until I looked at the supplemental data where I found the supplemental chart showing marker gene expression in the different clusters. The authors should mention how they did this in the main body of the text. I'm also guessing that the 'marker genes' for each population were discovered by others and these people and their work should be acknowledged, even if the data are publicly available.

Author response: In the main text of the revised manuscript, we have now incorporated a

description about how we assigned clusters to cell types and have cited the publications from which we obtained the marker gene information. In addition, the marker genes are now shown as an Expanded View figure, which offers a greatly enhanced online accessibility.

Figure 3B - I'm not sure where they came up with the pathway of cross-regulation between the four HSD-deactivated TFs they show in B. They should spell out where this pathway comes from or they should remove it.

Author response: In the initial submission, the cross-regulation shown in Figure 3B was extracted from the single-cell gene regulatory network. The network inference considers both gene co-expression patterns and TF binding motif enrichments. We acknowledge that the inferred interactions among these TFs haven't been validated except for the activation of Xbp1 by CrebA (Johnson et al., 2020). Therefore, we have decided to remove this panel and the related statements from the revised manuscript as suggested.

What is CrebA-lacZ? The authors need to explain what this is - I had to look it up but it is essentially a reporter insertion line for assaying CrebA expression (and the authors should give some caveats regarding using this line since it may not fully report on all aspects of CrebA expression, given it's a P-element insertion line that disrupts CrebA function).

Author response: We fully agree with this reviewer on the assessment on the use of this line. This line is the insertion line *l(3)3576* from the Allan Spradling enhancer-trap collection, which was characterized by Rose RE et al. (Genetics 1997) and Andrew DJ et al. (Development 1997). The insertion chromosome specifically expresses β -gal in majority of the tissues that normally express CrebA, including follicle cells at vitellogenic stages; however, it disrupts the expression of CrebA from this locus. Thus, this reporter line can be used to evaluate the transcription pattern of CrebA, for example, to validate the reduced transcription of CrebA in follicle cells from females subjected to a high-sucrose diet. Now we have incorporated this information in the revised manuscript. Importantly, as suggested by this reviewer, we added a caveat in the legend to the revised Figure 3C, noting that this line may not fully report all aspects of CrebA expression.

Figure 3D is not so convincing because not only is the green CrebA staining reduced with HSD, so is the purple DAPI nuclear staining and the DAPI looks to be reduced to a similar level as CrebA.

Author response: We thank this reviewer for raising this concern. Because the HSD females produced very few egg chambers beyond stage 8, we had a very limited number of good-quality images of egg chambers at stage 9 for the initial submission. This is despite the fact that all the samples in these experiments were prepared side by side, and their images were taken in a single operation cycle with consistent microscopic settings. To better illustrate the main conclusion of these results, we now present low-magnification images with more egg chambers, showing more clearly both the comparable DAPI intensities and differences in the CrebA signal between two diet conditions. In addition, we have performed a Western blot analysis shown in Figure 3D. Together, these results demonstrate a decrease of CrebA protein in ovaries of HSD females.

Figure 3 - The investigators need to provide more detail on how they found the "regulons" for each transcription factor they discuss. Again in figure S3 - please explain how the "regulons" were discovered and what is being plotted.

Author response: We have rewritten the related texts. Briefly, we used the bioinformatics tool SCENIC (van de Sande et al., 2020) to infer the single-cell transcription regulatory network. This method first identifies co-expression modules between TFs and candidate target genes from the scRNA-seq data. Then the binding potential of TF motifs on the candidate target genes is considered to prune the target gene lists and the modules. The refined modules are referred to as regulons. The regulon's activity is scored by assessing the expression ranks of the target genes in the transcriptome of each cell, and this score is binarized by fitting a Gaussian model for a better visual effect and shown in Figures 3A and EV3.

Figure 4F - why does the VM with the CrebA knockout look more electron dense than in the control - Can the authors show a higher mag view? Might help shed light on the differences and help the reader see them a little more easily.

Author response: We thank this reviewer for pointing out this observation. We have checked this particular aspect of our results across all of our available data. By comparing different samples from side-by-side experiments and using the oocyte content as a reference, we are inclined to believe that the difference in the electron density of vitelline membrane is likely real. In fact, vitelline membrane from egg chambers expressing the dominant negative InR in follicle cells also appeared thinner and denser, and CrebA overexpression appeared to restore both effects. While this electron density phenotype is potentially interesting with the underlying process warranting further investigation, we have decided not to delve into this aspect in the current manuscript to avoid potential confusion. We thank this reviewer for the expert evaluation of our results.

Figure 6A. Investigators need to acknowledge where the public ChIP-seq data for FoxO on the CrebA gene came from. Again, somebody did that work, even if it is publicly available.

Author response: We have incorporated a citation of the original article and the dataset for this ChIP-seq data in the revised manuscript.

Also, please show the actual sequence of the two FoxO motifs. As a reader, I would like to see how well the sites in the CrebA upstream region match the defined consensus binding motif for FoxO and I don't want to have to look it up by downloading the CrebA enhancer region.

Author response: We have included the sequences of the two sites alongside the consensus motif in the revised Figure 6B.

Bottom of page 12. The references for CrebA as a master regulator of the secretory machinery in embryonic and larval tissues are incorrect. Please remove the ones that are there and instead use: Abrams and Andrew, 2005; Fox et al., 2010; Johnson et al., 2020). Figure 7. The homology between CrebA and the Creb3 family has been pointed out previously and extends into a region N-terminal to the bZIP DNA binding domain - the N-terminal region distinguishes this class of molecules from other Crebs. The authors should probably recognize

the work that initially showed the extended homology and its importance in Creb3L function (Barbosa et al., 2013).

Author response: We thank this reviewer for pointing out these issues. We have incorporated the correct references. In addition, during this revision process we became aware of a recent publication on the role of CrebA in regulating transcription and the secretory pathway (Jackson et al., 2025). We regard this paper as another important reference, which is also cited in our revised manuscript.

Figure S8 A - the *tj-Gal4*>+ embryo shown is a late stage compared to the "freshly laid" *tj-Gal4*>CrebARN*Ai* embryo described. The authors should find and show a better stage comparison to match the CrebA knockdown embryo.

Author response: We thank this reviewer for making this suggestion. We have replaced the image of the control embryo in the revised manuscript.

Page 8, the description of what they used to monitor Yp1 expression is not quite right - it needs to be reworded - change "expresses" to "is" should read: which is a large genomic clone that includes the Yp1 gene fused to green fluorescent protein tag".

Author response: We have made the suggested rewording. We thank the reviewer for providing detailed and constructive suggestions that have led to a significant improvement of our manuscript.

Dear Dr. He,

Thank you for the submission of your revised manuscript to our editorial offices. I have now received the reports from the three referees that I asked to re-evaluate the study, you will find below. As you will see, the referees now fully support publication of your study in EMBO reports. Referee #3 has a suggestion to improve the manuscript, I ask you to address in a final revised manuscript. Please also provide a final p-b-p-response to this referee point and the editorial requests below.

Editorial requests:

- Please provide a final title with not more than 100 Characters including spaces.
- Please provide a final abstract with not more than 175 words.
- Please order the manuscript sections like this, using only these names:
Title page - Abstract - Keywords - Introduction - Results - Discussion - Methods - Data availability section - Acknowledgements - Disclosure and Competing Interests Statement - References - Figure legends - Expanded View Figure Legends
- We now use CRediT to specify the contributions of each author in the journal submission system. CRediT replaces the author contribution section. Please use the free text box to provide more detailed descriptions and do NOT provide your final manuscript text file with an author contributions section. See also our guide to authors:
<https://www.embopress.org/page/journal/14693178/authorguide#authorshipguidelines>
- Please use our reference format (using et al. for references with more than 10 authors):
<http://www.embopress.org/page/journal/14693178/authorguide#referencesformat>
- Please add scale bars of similar style and thickness to all the images, using clearly visible black or white bars (depending on the background), also in the Appendix. Please place these in the lower right corner of the images themselves. Please do not write on or near the bars in the image but define the size only in the respective figure legend. Presently, some scale bars are hard to see or have text nearby. Please check.
- Please check again that the number "n" for how many independent experiments were performed, their nature (biological versus technical replicates), the bars and error bars (e.g. SEM, SD) and the test used to calculate p-values is indicated in the respective figure legends (main and Appendix figures). Please also check that all the p-values are explained in the legend, and that these fit to those shown in the figure. Please provide statistical testing where applicable. Please avoid the phrase 'independent experiment' but clearly state if these were biological or technical replicates. Please also indicate (e.g. with n.s.) if testing was performed, but the differences are not significant. In case n=2, please show the data as separate datapoints without error bars and statistics. See also:
<http://www.embopress.org/page/journal/14693178/authorguide#statisticalanalysis>

If n<5, please show single datapoints for diagrams. Moreover:

Please define the annotated p values *****/**/* as well as provide the exact p-values for the same in the legend of figure 1D, G, 4A, F, 5A, B, D, 6C, D, F; 7C, E, G, H; S2C, S4, as appropriate.

- Please note that the exact p values are not provided in the legends of figures 4C, 5C, S1 A-D
- Please indicate the statistical test used for data analysis in the legends of figures 3F, 5C, S1 A-D
- Please note that the error bars are not defined in the legends of figures 1D, G; 3G, 4A, F; 5A, B, D, I; 6C, D, F; 7C, E, G, H, J; EV1 A-C.
- Please note that the VM, CH, OC are not defined in the legend of figure 5G, H. This needs to be rectified.
- Please remove the author names from the Appendix title page. Please just state 'Appendix for ...' followed by the final title of the manuscript (see above). However, please add page numbers to the Appendix and add a detailed Table of Contents (TOC) listing each Appendix item with page numbers. Moreover, please move the Appendix legends below each item. I think this is more comprehensible.
- Please add the primer information provided in Table S1 to the Reagents & Tools Table and add appropriate callouts to the Methods section. Then, please remove the table from the Appendix file. Please also remove the instructions from the final R&T Table.

In addition, I would need from you uploaded separately (please remove this from the manuscript text file):

- a short, two-sentence summary of the manuscript (not more than 35 words).
- two to four short (!) bullet points highlighting the key findings of your study (two lines each).
- a schematic summary figure as separate file that provides a sketch of the major findings (not a data image) in jpeg or tiff format

(with the exact width of 550 pixels and a height of not more than 400 pixels) that can be used as a visual synopsis on our website.

I look forward to seeing a new revised version of your manuscript as soon as possible.

Best,

Referee #1:

My previous concerns have been satisfactorily addressed in the revised version of the manuscript. I now recommend publication.

Referee #2:

I'm grateful to the authors, who have addressed my previous comments adequately.

Referee #3:

The authors did a stellar job addressing the issues raised by all three reviewers and the paper is much stronger for them having done so. I have only one suggestion - In figure 3, the authors are using the same labels to the left of panel B to work for panel E. I realize this is for space reasons, but it might be a bit better to have independent labels for both panels.

November 24, 2025

EMBO reports

Dear Editors,

We would like to respond to the referee's point and the editorial requests as below.

Referee #3: The authors did a stellar job addressing the issues raised by all three reviewers and the paper is much stronger for them having done so. I have only one suggestion - In figure 3, the authors are using the same labels to the left of panel B to work for panel E. I realize this is for space reasons, but it might be a bit better to have independent labels for both panels.

Authors: We have added the y-axis labels to the left of Figure 3E. Thank the referee for this suggestion.

Editorial requests:

- Please provide a final title with not more than 100 Characters including spaces.

Authors: We have modified the title.

- Please provide a final abstract with not more than 175 words.

Authors: We have modified the abstract.

- Please order the manuscript sections like this, using only these names:

Title page - Abstract - Keywords - Introduction - Results - Discussion - Methods - Data availability section - Acknowledgements - Disclosure and Competing Interests Statement - References - Figure legends - Expanded View Figure Legends

Authors: We have modified the manuscript sections according to this guidance.

- We now use CRediT to specify the contributions of each author in the journal submission system. CRediT replaces the author contribution section. Please use the free text box to provide more detailed descriptions and do NOT provide your final manuscript text file with an author contributions section. See also our guide to authors: <https://www.embopress.org/page/journal/14693178/authorguide#authorshipguidelines>

Authors: We have removed the author contributions section from the manuscript.

- Please use our reference format (using et al. for references with more than 10 authors):

Authors: We have modified the references.

- Please add scale bars of similar style and thickness to all the images, using clearly visible black or white bars (depending on the background), also in the Appendix.

Please place these in the lower right corner of the images themselves. Please do not write on or near the bars in the image but define the size only in the respective figure legend. Presently, some scale bars are hard to see or have text nearby. Please check.

Authors: We have adjusted the scale bars according to the guidance. The files of Figures 1, 3, 6 and EV5 have been replaced.

- Please check again that the number "n" for how many independent experiments were performed, their nature (biological versus technical replicates), the bars and error bars (e.g. SEM, SD) and the test used to calculate p-values is indicated in the respective figure legends (main and Appendix figures). Please also check that all the p-values are explained in the legend, and that these fit to those shown in the figure. Please provide statistical testing where applicable. Please avoid the phrase 'independent experiment' but clearly state if these were biological or technical replicates. Please also indicate (e.g. with n.s.) if testing was performed, but the differences are not significant. In case n=2, please show the data as separate datapoints without error bars and statistics. See also:

<http://www.embopress.org/page/journal/14693178/authorguide#statisticalanalysis>. If n<5, please show single datapoints for diagrams.

Authors: We have double checked all the reported statistics and made necessary changes according to the guidance.

Moreover: Please define the annotated p values ****/***/**/* as well as provide the exact p-values for the same in the legend of figure 1D, G, 4A, F, 5A, B, D, 6C, D, F; 7C, E, G, H; S2C, S4, as appropriate.

Authors: We have now provided all the exact p values together with their annotations in the figures.

- Please note that the exact p values are not provided in the legends of figures 4C, 5C, S1 A-D

Authors: We used Pearson's correlation tests for Figures 4C, 5C and S1A-D. The exact p values were extremely small. Now this information has been added to their legends.

- Please indicate the statistical test used for data analysis in the legends of figures 3F, 5C, S1 A-D

Authors: We used DESeq2 to perform the Wald test for Figure 3F. We used Pearson's correlation tests for Figures 4C, 5C and S1A-D. Now this information has been added to their legends.

- Please note that the error bars are not defined in the legends of figures 1D, G; 3G, 4A, F; 5A, B, D, I; 6C, D, F; 7C, E, G, H, J; EV1 A-C.

Authors: We have added the error bar definitions to the legends of all the figures with error bars.

- Please note that the VM, CH, OC are not defined in the legend of figure 5G, H. This needs to be rectified.

Authors: We have added this information to the legends of Figures 4F and 5G-H.

- Please remove the author names from the Appendix title page. Please just state 'Appendix for ...' followed by the final title of the manuscript (see above). However, please add page numbers to the Appendix and add a detailed Table of Contents (TOC) listing each Appendix item with page numbers. Moreover, please move the Appendix legends below each item. I think this is more comprehensible.

Authors: We have made the required changes to the Appendix.

- Please add the primer information provided in Table S1 to the Reagents & Tools Table and add appropriate callouts to the Methods section. Then, please remove the table from the Appendix file. Please also remove the instructions from the final R&T Table.

Authors: We have made the required changes to the Appendix, the Reagents & Tools Table, and the related in-text citation.

In addition, I would need from you uploaded separately (please remove this from the manuscript text file):

- a short, two-sentence summary of the manuscript (not more than 35 words).

Authors: Here is our summary:

Ovarian follicle cells undergo a stage-specific surge of insulin receptor activity, which is required for vitellogenesis and successful oogenesis.

We also include this information in a separate file uploaded into the submission system.

- two to four short (!) bullet points highlighting the key findings of your study (two lines each).

Authors: Here is our highlighting points:

- **Insulin receptor activity in *Drosophila* follicle cells exhibits a temporary surge during vitellogenesis**
- **CrebA transcriptionally activates yolk and vitelline membrane proteins in vitellogenic follicle cells**
- **Insulin receptor activity loss or high-sucrose diet suppresses CrebA expression through an increased FoxO activity**
- **The FOXO1-CREB3L2 regulatory module is the human counterpart in granulosa cells**

We also include this information in a separate file uploaded into the submission system.

- a schematic summary figure as separate file that provides a sketch of the major findings (not a data image) in jpeg or tiff format (with the exact width of 550 pixels

and a height of not more than 400 pixels) that can be used as a visual synopsis on our website.

Authors: We have uploaded a synopsis image to the submission system.

Best Regards,

The Authors

Feng He
Zhejiang University
School of Medicine
310058
China

Dear Dr. He,

I am very pleased to accept your manuscript for publication in the next available issue of EMBO reports. Thank you for your contribution to our journal.

You may qualify for financial assistance for your publication charges - either via a Springer Nature fully open access agreement or an EMBO initiative. Check your eligibility: <https://link.springer.com/journal/44319/how-to-publish-with-us>

Yours sincerely,

>>> Please note that it is EMBO Reports policy for the transcript of the editorial process (containing referee reports and your response letter) to be published as an online supplement to each paper. If you do NOT want this, you will need to inform the Editorial Office via email immediately. More information is available here: <https://link.springer.com/partners/embo-press/editorial-policies#Peer%20review>